# Efficient Symmetric Norm Regression via Linear Sketching[*]

**Zhao Song**
University of Washington
magic.linuxkde@gmail.com

**Ruosong Wang**
Carnegie Mellon University
ruosongw@andrew.cmu.edu

**Lin F. Yang**
University of California, Los Angeles
linyang@ee.ucla.edu

**Hongyang Zhang**
Toyota Technological Institute at Chicago
hongyanz@ttic.edu

**Peilin Zhong**
Columbia University
pz2225@columbia.edu

## Abstract

We provide efficient algorithms for overconstrained linear regression problems with size $n \times d$ when the loss function is a symmetric norm (a norm invariant under sign-flips and coordinate-permutations). An important class of symmetric norms are Orlicz norms, where for a function $G$ and a vector $y \in \mathbb{R}^n$, the corresponding Orlicz norm $\|y\|_G$ is defined as the unique value $\alpha$ such that $\sum_{i=1}^{n} G(|y_i|/\alpha) = 1$. When the loss function is an Orlicz norm, our algorithm produces a $(1 + \varepsilon)$-approximate solution for an arbitrarily small constant $\varepsilon > 0$ in input-sparsity time, improving over the previously best-known algorithm which produces a $d \cdot \operatorname{polylog} n$-approximate solution. When the loss function is a general symmetric norm, our algorithm produces a $\sqrt{d} \cdot \operatorname{polylog} n \cdot \operatorname{mmc}(\ell)$-approximate solution in input-sparsity time, where $\operatorname{mmc}(\ell)$ is a quantity related to the symmetric norm under consideration. To the best of our knowledge, this is the first input-sparsity time algorithm with provable guarantees for the general class of symmetric norm regression problem. Our results shed light on resolving the universal sketching problem for linear regression, and the techniques might be of independent interest to numerical linear algebra problems more broadly.

## 1 Introduction

Linear regression is a fundamental problem in machine learning. For a data matrix $A \in \mathbb{R}^{n \times d}$ and a response vector $b \in \mathbb{R}^n$ with $n \gg d$, the overconstrained linear regression problem can be formulated as solving the following optimization problem:

$$\min_{x \in \mathbb{R}^d} \mathcal{L}(Ax - b), \tag{1}$$

where $\mathcal{L} : \mathbb{R}^n \to \mathbb{R}$ is a loss function. Via the technique of linear sketching, we have witnessed many remarkable speedups for linear regression for a wide range of loss functions. Such technique involves designing a sketching matrix $S \in \mathbb{R}^{r \times n}$, and showing that by solving a linear regression instance on the data matrix $SA$ and the response vector $Sb$, which is usually much smaller in size, one can obtain

---

[*]All authors contribute equally.

Table 1: $M$-estimators

| HUBER | $\begin{cases} x^2/2 & |x| \le c \\ c(|x| - c/2) & |x| > c \end{cases}$ |
|---|---|
| $\ell_1 - \ell_2$ | $2(\sqrt{1 + x^2/2} - 1)$ |
| "FAIR" | $c^2\left(|x|/c - \log(1 + |x|/c)\right)$ |

an approximate solution to the original linear regression instance in (1). Sarlós showed in [29] that by taking $S$ as a Fast Johnson-Lindenstrauss Transform matrix [1], one can obtain $(1 + \varepsilon)$-approximate solutions to the least square regression problem $(\mathcal{L}(y) = \|y\|_2^2)$ in $O(nd \log n + \mathrm{poly}(d/\varepsilon))$ time. The running time was later improved to $O(\mathrm{nnz}(A) + \mathrm{poly}(d/\varepsilon))$ [12, 26, 28, 23, 15]. Here $\mathrm{nnz}(A)$ is the number of non-zero entries in the data matrix $A$, which could be much smaller than $nd$ for sparse data matrices. This technique was later generalized to other loss functions. By now, we have $\widetilde{O}(\mathrm{nnz}(A) + \mathrm{poly}(d/\varepsilon))$ time algorithms for $\ell_p$ norms $(\mathcal{L}(y) = \|y\|_p^p)$ [18, 26, 35, 16, 32], the quantile loss function [36], $M$-estimators [14, 13] and the Tukey loss function [11].

Despite we have successfully applied the technique of linear sketching to many different loss functions, ideally, it would be more desirable to design algorithms that work for a wide range of loss functions, instead of designing a new sketching algorithm for every specific loss function. Naturally, this leads to the following problem, which is the linear regression version of the universal sketching problem[2] studied in streaming algorithms [10, 9]. We note that similar problems are also asked and studied for various algorithmic tasks, including principal component analysis [31], sampling [21], approximate nearest neighbor search [4, 3], discrepancy [17, 8], sparse recovery [27] and mean estimation with statistical queries [19, 22].

**Problem 1.** *Is it possible to design sketching algorithms for linear regression, that work for a wide range of loss functions?*

Prior to our work, [14, 13] studied this problem in terms of $M$-estimators, where the loss function employs the form $\mathcal{L}(y) = \sum_{i=1}^{n} G(y_i)$ for some function $G$. See Table 1 for a list of $M$-estimators. However, much less is known for the case where the loss function $\mathcal{L}(\cdot)$ is a norm, except for $\ell_p$ norms. Recently, Andoni et al. [2] tackle Problem 1 for Orlicz norms, which can be seen as a scale-invariant version of $M$-estimators. For a function $G$ and a vector $y \in \mathbb{R}^n$ with $y \ne 0$, the corresponding Orlicz norm $\|y\|_G$ is defined as the unique value $\alpha$ such that

$$\sum_{i=1}^{n} G(|y_i|/\alpha) = 1. \tag{2}$$

When $y = 0$, we define $\|y\|_G$ to be 0. Note that Orlicz norms include $\ell_p$ norms as special cases, by taking $G(z) = |z|^p$ for some $p \ge 1$. Under certain assumptions on the function $G$, [2] obtains the first input-sparsity time algorithm for solving Orlicz norm regression. More precisely, in $\widetilde{O}(\mathrm{nnz}(A) + \mathrm{poly}(d \log n))$ time, their algorithm obtains a solution $\widehat{x} \in \mathbb{R}^d$ such that $\|A\widehat{x} - b\|_G \le d \cdot \mathrm{polylog}\, n \cdot \min_{x \in \mathbb{R}^d} \|Ax - b\|_G$.

There are two natural problems left open by the work of [2]. First, the algorithm in [2] has approximation ratio as large as $d \cdot \mathrm{polylog}\, n$. Although this result is interesting from a theoretical point of view, such a large approximation ratio is prohibitive for machine learning applications in practice. Is it possible to obtain an algorithm that runs in $\widetilde{O}(\mathrm{nnz}(A) + \mathrm{poly}(d/\varepsilon))$ time, with approximation ratio $1 + \varepsilon$, for arbitrarily small $\varepsilon$, similar to the case of $\ell_p$ norms? Moreover, although Orlicz norm includes a wide range of norms, many other important norms, e.g., top-$k$ norms (the sum of absolute values of the leading $k$ coordinates of a vector), max-mix of $\ell_p$ norms (e.g. $\max\{\|x\|_2, c\|x\|_1\}$ for some $c > 0$), and sum-mix of $\ell_p$ norms (e.g. $\|x\|_2 + c\|x\|_1$ for some $c > 0$), are not Orlicz norms. More complicated examples include the $k$-support norm [5] and the box-norm [25], which have found applications in sparse recovery. In light of Problem 1, it is natural to ask whether it is possible to apply the technique of linear sketching to a broader class of norms. In this paper, we obtain affirmative answers to both problems, and make progress towards finally resolving Problem 1.

**Notations.** We use $\widetilde{O}(f)$ to denote $f \, \mathrm{polylog}\, f$. For a matrix $A \in \mathbb{R}^{n \times d}$, we use $A_i \in \mathbb{R}^d$ to denote its $i$-th row, viewed as a column vector. For $n$ real numbers $x_1, x_2, \ldots, x_n$, we define

$\mathrm{diag}(x_1, x_2, \ldots, x_n) \in \mathbb{R}^{n \times n}$ to be the diagonal matrix where the $i$-th diagonal entry is $x_i$. For a vector $x \in \mathbb{R}^n$ and $p \geq 1$, we use $\|x\|_p$ to denote its $\ell_p$ norm, and $\|x\|_0$ to denote its $\ell_0$ norm, i.e., the number of non-zero entries in $x$. For two vectors $x, y \in \mathbb{R}^n$, we use $\langle x, y \rangle$ to denote their inner product. For any $n > 0$, we use $[n]$ to denote the set $\{1, 2, \ldots, n\}$. For $0 \leq p \leq 1$, we define $\mathrm{Ber}(p)$ to be the Bernoulli distribution with parameter $p$. We use $\mathbb{S}^{n-1}$ to denote the unit $\ell_2$ sphere in $\mathbb{R}^n$, i.e., $\mathbb{S}^{n-1} = \{x \in \mathbb{R}^n \mid \|x\|_2 = 1\}$. We use $\mathbb{R}_{\geq 0}$ to denote the set of all non-negative real numbers, i.e., $\mathbb{R}_{\geq 0} = \{x \in \mathbb{R} \mid x \geq 0\}$.

## 1.1 Our Contributions

**Algorithm for Orlicz Norms.** Our first contribution is a unified algorithm which produces $(1+\varepsilon)$-approximate solutions to the linear regression problem in (1), when the loss function $\mathcal{L}(\cdot)$ is an Orlicz norm. Before introducing our results, we first give our assumptions on the function $G$ which appeared in (2).

**Assumption 1.** *We assume the function $G : \mathbb{R} \to \mathbb{R}_{\geq 0}$ satisfies the following properties:*

1. *$G$ is a strictly increasing convex function on $[0, \infty)$;*

2. *$G(0) = 0$, and for all $x \in \mathbb{R}$, $G(x) = G(-x)$;*

3. *There exists some $C_G > 0$, such that for all $0 < x < y$, $G(y)/G(x) \leq C_G(y/x)^2$.*

The first two conditions in Assumption 1 are necessary to make sure the corresponding Orlicz norm $\|\cdot\|_G$ is indeed a norm, and the third condition requires the function $G$ to have at most quadratic growth, which can be satisfied by all $M$-estimators in Table 1 and is also required by prior work [2]. Notice that our assumptions are weaker than those in [2]. In [2], it is further required that $G(x)$ is a linear function when $x > 1$, and $G$ is twice differentiable on an interval $(0, \delta_G)$ for some $\delta_G > 0$. Given our assumptions on $G$, our main theorem is summarized as follows.

**Theorem 1.** *For a function $G$ that satisfies Assumption 1, there exists an algorithm that, on any input $A \in \mathbb{R}^{n \times d}$ and $b \in \mathbb{R}^n$, finds a vector $x^*$ in time $\widetilde{O}(\mathrm{nnz}(A) + \mathrm{poly}(d/\varepsilon))$, such that with probability at least $0.9$, $\|Ax^* - b\|_G \leq (1+\varepsilon) \min_{x \in \mathbb{R}^d} \|Ax - b\|_G$.*

To the best of our knowledge, this is the first input-sparsity time algorithm with $(1+\varepsilon)$-approximation guarantee, that goes beyond $\ell_p$ norms, the quantile loss function, and $M$-estimators. See Table 2 for a more comprehensive comparison with previous results.

**Algorithm for Symmetric Norms.** We further study the case when the loss function $\mathcal{L}(\cdot)$ is a symmetric norm. Symmetric norm is a more general class of norms, which includes all norms that are invariant under sign-flips and coordinate-permutations. Formally, we define symmetric norms as follow.

**Definition 1.** A norm $\|\cdot\|_\ell$ is called a *symmetric norm*, if $\|(y_1, y_2, \ldots, y_n)\|_\ell = \|(s_1 y_{\sigma_1}, s_2 y_{\sigma_2}, \ldots, s_n y_{\sigma_n})\|_\ell$ for any permutation $\sigma$ and any assignment of $s_i \in \{-1, 1\}$.

Symmetric norm includes $\ell_p$ norms and Orlicz norms as special cases. It also includes all examples provided in the introduction, i.e., top-$k$ norms, max-mix of $\ell_p$ norms, sum-mix of $\ell_p$ norms, the $k$-support norm [5] and the box-norm [25], as special cases. Understanding this general set of loss functions can be seen as a preliminary step to resolve Problem 1. Our main result for symmetric norm regression is summarized in the following theorem.

**Theorem 2.** *Given a symmetric norm $\|\cdot\|_\ell$, there exists an algorithm that, on any input $A \in \mathbb{R}^{n \times d}$ and $b \in \mathbb{R}^n$, finds a vector $x^*$ in time $\widetilde{O}(\mathrm{nnz}(A) + \mathrm{poly}(d))$, such that with probability at least $0.9$, $\|Ax^* - b\|_\ell \leq \sqrt{d} \cdot \mathrm{polylog}\, n \cdot \mathrm{mmc}(\ell) \cdot \min_{x \in \mathbb{R}^d} \|Ax - b\|_\ell$.*

In the above theorem, $\mathrm{mmc}(\ell)$ is a characteristic of the symmetric norm $\|\cdot\|_\ell$, which has been proven to be essential in streaming algorithms for symmetric norms [7]. See Definition 7 for the formal definition of $\mathrm{mmc}(\ell)$, and Section 3 for more details about $\mathrm{mmc}(\ell)$. In particular, for $\ell_p$ norms with $p \leq 2$, top-$k$ norms with $k \geq n/\mathrm{polylog}\, n$, max-mix of $\ell_2$ norm and $\ell_1$ norm ($\max\{\|x\|_2, c\|x\|_1\}$ for some $c > 0$), sum-mix of $\ell_2$ norm and $\ell_1$ norm ($\|x\|_2 + c\|x\|_1$ for some $c > 0$), the $k$-support norm, and the box-norm, $\mathrm{mmc}(\ell)$ can all be upper bounded by $\mathrm{polylog}\, n$, which implies our algorithm has approximation ratio $\sqrt{d} \cdot \mathrm{polylog}\, n$ for all these norms. This clearly demonstrates the generality of our algorithm.

Table 2: Comparison among input-sparsity time linear regression algorithms

| Reference | Loss Function | Approximation Ratio |
|---|---|---|
| [18, 26, 35, 16, 32] | $\ell_p$ norms | $1 + \varepsilon$ |
| [36] | Quantile loss function | $1 + \varepsilon$ |
| [14, 13] | $M$-estimators | $1 + \varepsilon$ |
| [2] | Orlicz norms | $d \cdot \mathrm{polylog}\, n$ |
| **Theorem 1** | Orlicz norms | $1 + \varepsilon$ |
| **Theorem 2** | Symmetric norms | $\sqrt{d} \cdot \mathrm{polylog}\, n \cdot \mathrm{mmc}(\ell)$ |

**Empirical Evaluation.**    In Section E of the supplementary material, we test our algorithms on real datasets. Our empirical results quite clearly demonstrate the practicality of our methods.

## 1.2   Technical Overview

Similar to previous works on using linear sketching to speed up solving linear regression, our core technique is to provide efficient dimensionality reduction methods for Orlicz norms and general symmetric norms. In this section, we discuss the techniques behind our results.

**Row Sampling Algorithm for Orlicz Norms.**    Compared to prior work on Orlicz norm regression [2] which is based on random projection[3], our new algorithm is based on row sampling. For a given matrix $A \in \mathbb{R}^{n \times d}$, our goal is to output a *sparse* weight vector $w \in \mathbb{R}^n$ with at most $\mathrm{poly}(d \log n / \varepsilon)$ non-zero entries, such that with high probability, for all $x \in \mathbb{R}^d$,

$$(1 - \varepsilon)\|Ax - b\|_G \le \|Ax - b\|_{G,w} \le (1 + \varepsilon)\|Ax - b\|_G. \tag{3}$$

Here, for a weight vector $w \in \mathbb{R}^n$ and a vector $y \in \mathbb{R}^n$, the *weighted Orlicz norm* $\|y\|_{G,w}$ is defined as the unique value $\alpha$ such that $\sum_{i=1}^{n} w_i G(|y_i|/\alpha) = 1$. See Definition 4 for the formal definition of weighted Orlicz norm. To obtain a $(1 + \varepsilon)$-approximate solution to Orlicz norm regression, by (3), it suffices to solve

$$\min_{x \in \mathbb{R}^d} \|Ax - b\|_{G,w}. \tag{4}$$

Since the vector $w \in \mathbb{R}^n$ has at most $\mathrm{poly}(d \log n / \varepsilon)$ non-zero entries, and we can ignore all rows of $A$ with zero weights, there are at most $\mathrm{poly}(d \log n / \varepsilon)$ remaining rows in $A$ in the optimization problem in (4). Furthermore, as we show in Lemma 3, $\| \cdot \|_{G,w}$ is a seminorm, which implies we can solve the optimization problem in (4) in $\mathrm{poly}(d \log n / \varepsilon)$ time, by simply solving a convex program with size $\mathrm{poly}(d \log n / \varepsilon)$. Thus, we focus on how to obtain the weight vector $w \in \mathbb{R}^n$ in the remaining part. Furthermore, by taking $\overline{A}$ to be a matrix whose first $d$ columns are $A$ and last column is $b$, to satisfy (3), it suffices to find a weight vector $w$ such that for all $x \in \mathbb{R}^{d+1}$,

$$(1 - \varepsilon)\|\overline{A}x\|_G \le \|\overline{A}x\|_{G,w} \le (1 + \varepsilon)\|\overline{A}x\|_G. \tag{5}$$

Hence, we ignore the response vector $b$ in the remaining part of the discussion.

We obtain the weight vector $w$ via importance sampling. We compute a set of sampling probabilities $\{p_i\}_{i=1}^{n}$ for each row of the data matrix $A$, and sample the rows of $A$ according to these probabilities. The $i$-th entry of the weight vector $w$ is then set to be $w_i = 1/p_i$ with probability $p_i$ and $w_i = 0$ with probability $1 - p_i$. However, unlike $\ell_p$ norms, Orlicz norms are not "entry-wise" norms, and it is not even clear that such a sampling process gives an unbiased estimation. Our key insight here is that for a vector $Ax$ with unit Orlicz norm, if for all $x \in \mathbb{R}^d$,

$$(1 - \varepsilon) \sum_{i=1}^{n} G((Ax)_i) \le \sum_{i=1}^{n} w_i G((Ax)_i) \le (1 + \varepsilon) \sum_{i=1}^{n} G((Ax)_i), \tag{6}$$

then (5) holds, which follows from the convexity of the function $G$. See Lemma 7 and its proof for more details. Therefore, it remains to develop a way to define and calculate $\{p_i\}_{i=1}^{n}$, such that the total number of sampled rows is small.

Our method for defining and computing sampling probabilities $p_i$ is inspired by row sampling algorithms for $\ell_p$ norms [18]. Here, the key is to obtain an upper bound on the contribution of each entry to the summation $\sum_{i=1}^{n} G((Ax)_i)$. Indeed, suppose for some vector $u \in \mathbb{R}^n$ such that $G(Ax)_i \le u_i$ for all $x \in \mathbb{R}^d$ with $\|Ax\|_G = 1$, we can then sample each row of $A$ with sampling probability proportional to $u_i$. Now, by standard concentration inequalities and a net argument, (6) holds with high probability. It remains to upper bound the total number of sampled rows, which is proportional to $\sum_{i=1}^{n} u_i$.

We use the case of $\ell_2$ norm, i.e., $G(x) = x^2$, as an example to illustrate our main ideas for choosing the vector $u \in \mathbb{R}^n$. Suppose $U \in \mathbb{R}^{n \times d}$ is an orthonormal basis matrix of the column space of $A$, then the *leverage score*[4] is defined to be the squared $\ell_2$ norm of each row of $U$. Indeed, leverage score gives an upper bound on the contribution of each row to $\|Ux\|_2^2$, since by Cauchy-Schwarz inequality, for each row $U_i$ of $U$, we have $\langle U_i, x \rangle^2 \le \|U_i\|_2^2 \|x\|_2^2 = \|U_i\|_2^2 \|Ux\|_2^2$, and thus we can set $u_i = \|U_i\|_2^2$. It is also clear that $\sum_{i=1}^{n} u_i = d$.

For general Orlicz norms, leverage scores are no longer upper bounds on $G((Ux)_i)$. Inspired by the role of orthonormal bases in the case of $\ell_2$ norm, we first define well-conditioned basis for general Orlicz norms as follow.

**Definition 2.** Let $\| \cdot \|_G$ be an Orlicz norm induced by a function $G$ which satisfies Assumption 1. We say $U \in \mathbb{R}^{n \times d}$ is a well-conditioned basis with condition number $\kappa_G = \kappa_G(U)$ if for all $x \in \mathbb{R}^d$, $\|x\|_2 \le \|Ux\|_G \le \kappa_G \|x\|_2$.

Given this definition, when $\|Ux\|_G = 1$, by Cauchy-Schwarz inequality and monotonicity of $G$, we can show that $G((Ux)_i) \le G(\|U_i\|_2 \|x\|_2) \le G(\|U_i\|_2 \|Ux\|_G) \le G(\|U_i\|_2)$. This also leads to our definition of Orlicz norm leverage scores.

**Definition 3.** Let $\| \cdot \|_G$ be an Orlicz norm induced by a function $G$ which satisfies Assumption 1. For a given matrix $A \in \mathbb{R}^{n \times d}$ and a well-conditioned basis $U$ of the column space of $A$, the *Orlicz norm leverage score* of the $i$-th row of $A$ is defined to be $G(\|U_i\|_2)$.

It remains to give an upper bound on the summation of Orlicz norm leverage scores of all rows. Unlike the $\ell_2$ norm, it is not immediately clear how to use the definition of well-conditioned basis to obtain such an upper bound for general Orlicz norms. To achieve this goal, we use a novel probabilistic argument. Suppose one takes $x$ to be a vector with i.i.d. Gaussian random variables. Then each entry of $Ux$ has the same distribution as $\|U_i\|_2 \cdot g_i$, where $\{g_i\}_{i=1}^{n}$ is a set of standard Gaussian random variables. Thus, with constant probability, $\sum_{i=1}^{n} G((Ux)_i)$ is an upper bound on the summation of Orlicz norm leverage scores. Furthermore, by the growth condition of the function $G$, we have $\sum_{i=1}^{n} G((Ux)_i) \le C_G \|Ux\|_G^2$. Now by Definition 2, $\|Ux\|_G \le \kappa_G \|x\|_2$, and $\|x\|_2 \le O(\sqrt{d})$ with constant probability by tail inequalities of Gaussian random variables. This implies an upper bound on the summation of Orlicz norm leverage scores. See Lemma 4 and its proof for more details.

Our approach for constructing well-conditioned bases is inspired by [30]. In Lemma 5, we show that given a subspace embedding $\Pi$ which embeds the column space of $A$ with Orlicz norm $\| \cdot \|_G$ into the $\ell_2$ space with distortion $\kappa$, then one can construct a well-conditioned basis with condition number $\kappa_G \le \kappa$. The running time is dominated by calculating $\Pi A$ and doing a QR-decomposition on $\Pi A$. To this end, we can use the oblivious subspace embedding for Orlicz norms in Corollary 12[5] to construct well-conditioned bases. The embedding in Corollary 12 has $O(d)$ rows and $\kappa = \text{poly}(d \log n)$, and calculating $\Pi A$ can be done in $\widetilde{O}(\text{nnz}(A) + \text{poly}(d))$ time. Using such an embedding to construct the well-conditioned basis, our row sampling algorithm produces a vector $w$ that satisfies (6) with $\|w\|_0 \le \text{poly}(d \log n / \varepsilon)$ in time $\widetilde{O}(\text{nnz}(A) + \text{poly}(d))$.

We would like to remark that our sampling algorithm still works if the third condition in Assumption 1 does not hold. In general, suppose the function $G : \mathbb{R} \to \mathbb{R}$ satisfies that for all $0 < x < y$, $G(y)/G(x) \le C_G(y/x)^p$, for the Orlicz norm induced by $G$, given a well-conditioned basis with condition number $\kappa_G$, our sampling algorithm returns a matrix with roughly $O((\sqrt{d}\kappa_G)^p \cdot d/\varepsilon^2)$ rows such that Theorem 1 holds. One may use the Löwner–John ellipsoid as the well-conditioned

basis (as in [18]) which has condition number $\kappa_G = \sqrt{d}$ for any norm. However, calculating the Löwner–John ellipsoid requires at least $O(nd^5)$ time. Moreover, our method described above fails when $p > 2$ since it requires an oblivious subspace embedding with $\mathrm{poly}(d)$ distortion, and it is known that such embedding does not exist when $p > 2$ [10]. Since we focus on input-sparsity time algorithms in this paper, we only consider the case that $p \leq 2$.

Finally, we would like to compare our sampling algorithm with that in [13]. First, the algorithm in [13] works for $M$-estimators, while we focus on Orlicz norms. Second, our definitions for Orlicz norm leverage score and well-conditioned basis, as given in Definition 2 and 3, are different from all previous works and are closely related to the Orlicz norm under consideration. The algorithm in [13], on the other hand, simply uses $\ell_p$ leverage scores. Under our definition, we can prove that the sum of leverage scores is bounded by $O(C_G d \kappa_G^2)$ (Lemma 4), whose proof requires a novel probabilistic argument. In contrast, the upper bound on sum of leverage scores in [13] is $O(\sqrt{nd})$ (Lemma 38 in [11]). Thus, the algorithm in [13] runs in an iterative manner since in each round the algorithm can merely reduce the dimension from $n$ to $O(\sqrt{nd})$, while our algorithm is one-shot.

**Oblivious Subspace Embeddings for Symmetric Norms.** To obtain a faster algorithm for linear regression when the loss function is a general symmetric norm, we show that there exists a distribution over embedding matrices, such that if $S$ is a random matrix drawn from that distribution, then for any $n \times d$ matrix $A$, with constant probability, for all $x \in \mathbb{R}^d$, $\|Ax\|_\ell \leq \|SAx\|_2 \leq \mathrm{poly}(d \log n) \cdot \mathrm{mmc}(\ell) \cdot \|Ax\|_\ell$. Moreover, the embedding matrix $S$ is *sparse*, and calculating $SA$ requires only $\widetilde{O}(\mathrm{nnz}(A) + \mathrm{poly}(d))$ time. Another favorable property of $S$ is that it is an *oblivious subspace embeeding*, meaning the distribution of $S$ does not depend on $A$. To achieve this goal, it is sufficient to construct a random diagonal matrix $D$ such that for any fixed vector $x \in \mathbb{R}^n$,

$$\Pr[\|Dx\|_2 \geq \Omega(1/\mathrm{poly}(d \log n)) \cdot \|x\|_\ell] \geq 1 - \exp(-\Omega(d \log n)), \tag{7}$$

and

$$\Pr[\|Dx\|_2 \leq \mathrm{poly}(d \log n) \cdot \mathrm{mmc}(\ell) \cdot \|x\|_\ell] \geq 1 - O(1/d). \tag{8}$$

Our construction is inspired by the sub-sampling technique in [20], which was used for sketching symmetric norms in data streams [7]. Throughout the discussion, we use $\xi^{(q)} \in \mathbb{R}^n$ to denote a vector with $q$ non-zero entries and each entry is $1/\sqrt{q}$. Let us start with a special case where the vector $x \in \mathbb{R}^n$ has $s$ non-zero entries and each non-zero entry is 1. It is easy to see $\|x\|_\ell = \sqrt{s}\|\xi^{(s)}\|_\ell$. Now consider a random diagonal matrix $D$ which corresponds to a sampling process, i.e., each diagonal entry is set to be 1 with probability $p$ and 0 with probability $1 - p$. Our goal is to show that $\sqrt{1/p}\|\xi^{(1/p)}\|_\ell \cdot \|Dx\|_2$ is a good estimator of $\|x\|_\ell$. If $p = \Theta(d \log n/s)$, then with probability at least $1 - \exp\left(-\Omega(d \log n)\right)$, $Dx$ will contain at least one non-zero entry from $x$, in which case (7) is satisfied. However, we do not know $s$ in advance. Thus, we use $t = O(\log n)$ different matrices $D_1, D_2, \ldots, D_t$, where $D_i$ has sampling probability $1/2^i$. Clearly at least one such $D_j$ can establish (7). For the upper bound part, if $p$ is much smaller than $1/s$, then $Dx$ will never contain a non-zero entry from $x$. Otherwise, in expectation $Dx$ will contain $ps$ non-zero entries, in which case our estimation will be roughly $\sqrt{s}\|\xi^{(1/p)}\|_\ell$, which can be upper bounded by $O(\log n \cdot \mathrm{mmc}(\ell) \cdot \sqrt{s}\|\xi^{(s)}\|_\ell)$. At this point, (8) follows from Markov's inequality. See Section C.5 for the formal argument, and Section 3 for a detailed discussion on $\mathrm{mmc}(\ell)$.

To generalize the above argument to general vectors, for a vector $x \in \mathbb{R}^n$, we conceptually partition its entries into $\Theta(\log n)$ groups, where the $i$-th group contains entries with magnitude in $[2^i, 2^{i+1})$. By averaging, at least one group of entries contributes at least $\Omega(1/\log n)$ fraction to the value of $\|x\|_\ell$. To establish (7), we apply the lower bound part of the argument in the previous paragraph to this "contributing" group. To establish (8), we apply the upper bound part of the argument to all groups, which will only induce an additional $O(\log n)$ factor in the approximation ratio, by triangle inequality.

Since our oblivious subspace embedding embeds a given symmetric norm into the $\ell_2$ space, in order to obtain an approximate solution to symmetric norm regression, we only need to solve a least squares regression instance with much smaller size. This is another advantage of our subspace embedding, since the least square regression problem is a well-studied problem in optimization and numerical linear algebra, for which many efficient algorithms are known, both in theory and in practice.

## 2 Linear Regression for Orlicz Norms

In this section, we introduce our results for Orlicz norm regression. We first give the definition of weighted Orlicz norm.

**Definition 4.** For a function $G$ that satisfies Assumption 1 and a weight vector $w \in \mathbb{R}^n$ such that $w_i \geq 0$ for all $i \in [n]$, for a vector $x \in \mathbb{R}^n$, if $\sum_{i=1}^n w_i \cdot |x_i| = 0$, then the weighted Orlicz norm $\|x\|_{G,w}$ is defined to be 0. Otherwise, the weighted Orlicz norm $\|x\|_{G,w}$ is defined as the unique value $\alpha > 0$ such that $\sum_{i=1}^n w_i G(|x_i|/\alpha) = 1$.

When $w_i = 1$ for all $i \in [n]$, we have $\|x\|_{G,w} = \|x\|_G$ where $\|x\|_G$ is the (unweighted) Orlicz norm. It is well known that $\|\cdot\|_G$ is a norm. We show in the following lemma that $\|\cdot\|_{G,w}$ is a seminorm.

**Lemma 3.** *For a function $G$ that satisfies Assumption 1 and a weight vector $w \in \mathbb{R}^n$ such that $w_i \geq 0$ for all $i \in [n]$, for all $x, y \in \mathbb{R}^n$, we have (i) $\|x\|_{G,w} \geq 0$, (ii) $\|x+y\|_{G,w} \leq \|x\|_{G,w} + \|y\|_{G,w}$, and (iii) $\|ax\|_{G,w} = |a| \cdot \|x\|_{G,w}$ for all $a \in \mathbb{R}$.*

**Leverage Scores and Well-Conditioned Bases for Orlicz Norms.** The following lemma establishes an upper bound on the summation of Orlicz norm leverage scores defined in Definition 3.

**Lemma 4.** *Let $\|\cdot\|_G$ be an Orlicz norm induced by a function $G$ which satisfies Assumption 1. Let $U \in \mathbb{R}^{n \times d}$ be a well-conditioned basis with condition number $\kappa_G$ as in Definition 2. Then we have $\sum_{i=1}^n G(\|U_i\|_2) \leq O(C_G d \kappa_G^2)$,*

Now we show that given a subspace embedding which embeds the column space of $A$ with Orlicz norm $\|\cdot\|_G$ into the $\ell_2$ space with distortion $\kappa$, then one can construct a well-conditioned basis with condition number $\kappa_G \leq \kappa$.

**Lemma 5.** *Let $\|\cdot\|_G$ be an Orlicz norm induced by a function $G$ which satisfies Assumption 1. For a given matrix $A \in \mathbb{R}^{n \times d}$ and an embedding matrix $\Pi \in \mathbb{R}^{s \times n}$, suppose for all $x \in \mathbb{R}^d$, $\|Ax\|_G \leq \|\Pi Ax\|_2 \leq \kappa \|Ax\|_G$. Let $Q \cdot R = \frac{1}{\kappa} \Pi A$ be a QR-decomposition of $\frac{1}{\kappa} \Pi A$. Then $AR^{-1}$ is a well-conditioned basis (see Definition 2) with $\kappa_G(AR^{-1}) \leq \kappa$.*

The following lemma shows how to estimate Orlicz norm leverage scores given a change of basis matrix $R \in \mathbb{R}^{d \times d}$, in $\widetilde{O}(\mathrm{nnz}(A) + \mathrm{poly}(d))$ time.

**Lemma 6.** *Let $\|\cdot\|_G$ be an Orlicz norm induced by a function $G$ which satisfies Assumption 1. For a given matrix $A \in \mathbb{R}^{n \times d}$ and $R \in \mathbb{R}^{d \times d}$, there exists an algorithm that outputs $\{u_i\}_{i=1}^n$ such that with probability at least 0.99, $u_i = \Theta(G(\|(AR^{-1})_i\|_2))$ for all $1 \leq i \leq n$. The algorithm runs in $\widetilde{O}(\mathrm{nnz}(A) + \mathrm{poly}(d))$ time.*

**The Row Sampling Algorithm.** Based on the notion of Orlicz norm leverage scores and well-conditioned bases, we design a row sampling algorithm for Orlicz norms.

**Lemma 7.** *Let $\|\cdot\|_G$ be an Orlicz norm induced by a function $G$ which satisfies Assumption 1. Let $U \in \mathbb{R}^{n \times d}$ be a well-conditioned basis with condition number $\kappa_G = \kappa_G(U)$ as in Definition 2. For sufficiently small $\varepsilon$ and $\delta$, and sufficiently large constant $C$, let $\{p_i\}_{i=1}^n$ be a set of sampling probabilities satisfying $p_i \geq \min\left\{1, C\left(\log(1/\delta) + d\log(1/\varepsilon)\right)\varepsilon^{-2}G\left(\|U_i\|_2\right)\right\}$. Let $w$ be a vector whose $i$-th entry is set to be $w_i = 1/p_i$ with probability $p_i$ and $w_i = 0$ with probability $1 - p_i$, then with probability at least $1 - \delta$, for all $x \in \mathbb{R}^d$, we have $(1-\varepsilon)\|Ux\|_G \leq \|Ux\|_{G,w} \leq (1+\varepsilon)\|Ux\|_G$.*

**Solving Linear Regression for Orlicz Norms.** Now we combine all ingredients to give an algorithm for Orlicz norm regression. We use $\overline{A} \in \mathbb{R}^{n \times (d+1)}$ to denote a matrix whose first $d$ columns are $A$ and the last column is $b$. The algorithm is described in Figure 1, and we prove its running time and correctness in Theorem 8. We assume we are given an embedding matrix $\Pi$, such that for all $x \in \mathbb{R}^{d+1}$, $\|\overline{A}x\|_G \leq \|\Pi\overline{A}x\|_2 \leq \kappa\|\overline{A}x\|_G$. The construction of $\Pi$ and the value $\kappa$ will be given in Corollary 12. In Section D.1 of the supplementary material, we use Theorem 8 and Corollary 12 to formally prove Theorem 1.

---

1. For the given embedding matrix $\Pi$, calculate $\Pi\overline{A}$ and invoke QR-decomposition on $\Pi\overline{A}/\kappa$ to obtain $Q \cdot R = \Pi\overline{A}/\kappa$.

2. Invoke Lemma 6 to obtain $\{u_i\}_{i=1}^n$ such that $u_i = \Theta(G(\|(\overline{A}R^{-1})_i\|_2))$.

3. For a sufficiently large constant $C$, let $\{p_i\}_{i=1}^n$ be a set of sampling probabilities with $p_i \geq \min\left\{1, C \cdot d \cdot \varepsilon^{-2}\log(1/\varepsilon) \cdot G\left(\|(\overline{A}R^{-1})_i\|_2\right)\right\}$, and $w$ be a vector whose $i$-th entry $w_i = 1/p_i$ with probability $p_i$ and $w_i = 0$ with probability $1 - p_i$.

4. Calculate $x^* = \operatorname{argmin}_{x\in\mathbb{R}^d}\|Ax-b\|_{G,w}$. Return $x^*$.

---

Figure 1: Algorithm for Orlicz norm regression

**Theorem 8.** *Let $\|\cdot\|_G$ be an Orlicz norm induced by a function $G$ which satisfies Assumption 1. Given an embedding matrix $\Pi$, such that for all $x \in \mathbb{R}^d$, $\|\overline{A}x\|_G \leq \|\Pi\overline{A}x\|_2 \leq \kappa\|\overline{A}x\|_G$, with probability at least $0.9$, the algorithm in Figure 1 outputs $x^* \in \mathbb{R}^d$ in time $\operatorname{poly}(d\kappa/\varepsilon) + \mathcal{T}_{\mathrm{QR}}(\Pi\overline{A})$, such that $\|Ax^* - b\|_G \leq (1+\varepsilon)\min_{x\in\mathbb{R}^d}\|Ax - b\|_G$. Here, $\mathcal{T}_{\mathrm{QR}}(\Pi\overline{A})$ is the running time for calculating $\Pi\overline{A}$ and invoking QR-decomposition on $\Pi\overline{A}$.*

## 3 Linear Regression for Symmetric Norms

In this section, we introduce SymSketch, a subspace embedding for symmetric norms.

**Definition of SymSketch.** We first formally define SymSketch. Due to space limitation, we give the definition of Gaussian embeddings, CountSketch embeddings and their compositions in Section C.1.1 of the supplementary material.

**Definition 5** (Symmetric Norm Sketch (SymSketch)). Let $t = \Theta(\log n)$. Let $\widetilde{D} \in \mathbb{R}^{n(t+1)\times n}$ be a matrix defined as $\widetilde{D} = \begin{bmatrix}(w_0 D_0)^\top & (w_1 D_1)^\top & \ldots & (w_t D_t)^\top\end{bmatrix}^\top$, where for each $i \in \{0,1,\ldots,t\}$, $D_i = \operatorname{diag}(z_{i,1}, z_{i,2}, \ldots, z_{i,n}) \in \mathbb{R}^{n\times n}$ and $z_{i,j} \sim \operatorname{Ber}(1/2^i)$ for each $j \in [n]$. Moreover, $w_i = \|(1,1,\ldots,1,0,\ldots,0)\|_\ell$ (there are $2^i$ 1s). Let $\Pi \in \mathbb{R}^{O(d)\times n(t+1)}$ be a composition of Gaussian embedding and CountSketch embedding (Definition 12) with $\varepsilon = 0.1$, and $S = \Pi\widetilde{D}$. We say $S \in \mathbb{R}^{O(d)\times n}$ is a SymSketch.

**Modulus of Concentration.** Now we give the definition of $\operatorname{mmc}(\ell)$ for a symmetric norm.

**Definition 6** ([7]). Let $\mathcal{X}$ denote the uniform distribution over $\mathbb{S}^{n-1}$. The *median* of a symmetric norm $\|\cdot\|_\ell$ is the unique value $M_\ell$ such that $\Pr_{x\sim\mathcal{X}}[\|x\|_\ell \geq M_\ell] \geq 1/2$ and $\Pr_{x\sim\mathcal{X}}[\|x\|_\ell \leq M_\ell] \geq 1/2$.

**Definition 7** ([7]). For a given symmetric norm $\|\cdot\|_\ell$, we define the *modulus of concentration* to be $\operatorname{mc}(\ell) = \max_{x\in\mathbb{S}^{n-1}}\|x\|_\ell/M_\ell$, and define the *maximum modulus of concentration* to be $\operatorname{mmc}(\ell) = \max_{k\in[n]}\operatorname{mc}(\ell^{(k)})$, where $\|\cdot\|_{\ell^{(k)}}$ is a norm on $\mathbb{R}^k$ which is defined to be $\|(x_1, x_2, \ldots, x_k)\|_{\ell^{(k)}} = \|(x_1, x_2, \ldots, x_k, 0, \ldots, 0)\|_\ell$.

It has been shown in [7] that $\operatorname{mmc}(\ell) = \Theta(n^{1/2-1/p})$ for $\ell_p$ norms when $p > 2$, $\operatorname{mmc}(\ell) = \Theta(1)$ for $\ell_p$ norms when $p \leq 2$, $\operatorname{mmc}(\ell) = \widetilde{\Theta}(\sqrt{n/k})$ for top-$k$ norms, and $\operatorname{mmc}(\ell) = O(\log n)$ for the $k$-support norm [5] and the box-norm [25]. We show that $\operatorname{mmc}(\ell)$ is upper bounded by $O(1)$ for max-mix of $\ell_2$ norm and $\ell_1$ norm and sum-mix of $\ell_2$ norm and $\ell_1$ norm.

**Lemma 9.** *For a real number $c > 0$, let $\|x\|_{\ell_a} = \|x\|_2 + c\|x\|_1$ and $\|x\|_{\ell_b} = \max\{\|x\|_2, c\|x\|_1\}$. We have $\operatorname{mmc}(\ell_a) = O(1)$ and $\operatorname{mmc}(\ell_b) = O(1)$.*

Moreover, we show that for an Orlicz norm $\|\cdot\|_G$ induced by a function $G$ which satisfies Assumption 1, $\operatorname{mmc}(\ell)$ is upper bounded by $O(\sqrt{C_G}\log n)$.

**Lemma 10.** *For an Orlicz norm $\|\cdot\|_G$ on $\mathbb{R}^n$ induced by a function $G$ which satisfies Assumption 1, $\operatorname{mmc}(\ell)$ is upper bounded by $O(\sqrt{C_G}\log n)$.*

**Subspace Embedding.** The following theorem shows that SymSketch is a subspace embedding.

**Theorem 11.** *Let $S \in \mathbb{R}^{O(d) \times n}$ be a* **SymSketch** *as defined in Definition 5. For a given matrix $A \in \mathbb{R}^{n \times d}$, with probability at least $0.9$, for all $x \in \mathbb{R}^d$,*

$$\Omega\left(1/(\sqrt{d} \cdot \log^3 n)\right) \cdot \|Ax\|_\ell \leq \|SAx\|_2 \leq O\left(\mathrm{mmc}(\ell) \cdot d^2 \cdot \log^{5/2} n\right) \cdot \|Ax\|_\ell.$$

*Furthermore, the running time of computing $SA$ is $\widetilde{O}(\mathrm{nnz}(A) + \mathrm{poly}(d))$.*

Combine Theorem 11 with Lemma 10, we have the following corollary.

**Corollary 12.** *Let $\|\cdot\|_G$ be an Orlicz norm induced by a function $G$ which satisfies Assumption 1. Let $S \in \mathbb{R}^{O(d) \times n}$ be a* **SymSketch** *as defined in Definition 5. For a given matrix $A \in \mathbb{R}^{n \times d}$, with probability at least $0.9$, for all $x \in \mathbb{R}^d$,*

$$\Omega\left(1/(\sqrt{d} \cdot \log^3 n)\right) \cdot \|Ax\|_\ell \leq \|SAx\|_2 \leq O\left(\sqrt{C_G} \cdot d^2 \cdot \log^{7/2} n\right) \cdot \|Ax\|_\ell.$$

*Furthermore, the running time of computing $SA$ is $\widetilde{O}(\mathrm{nnz}(A) + \mathrm{poly}(d))$.*

## 4 Conclusion

In this paper, we give efficient algorithms for solving the overconstrained linear regression problem, when the loss function is a symmetric norm. For the special case when the loss function is an Orlicz norm, our algorithm produces a $(1+\varepsilon)$-approximate solution in $\widetilde{O}(\mathrm{nnz}(A) + \mathrm{poly}(d/\varepsilon))$ time. When the loss function is a general symmetric norm, our algorithm produces a $\sqrt{d} \cdot \mathrm{polylog}\, n \cdot \mathrm{mmc}(\ell)$-approximate solution in $\widetilde{O}(\mathrm{nnz}(A) + \mathrm{poly}(d))$ time.

In light of Problem 1, there are a few interesting problems that remain open. Is that possible to design an algorithm that produces $(1 + \varepsilon)$-approximate solutions to the linear regression problem, when the loss function is a general symmetric norm? Furthermore, is that possible to use the technique of linear sketching to speed up the overconstrained linear regression problem, when the loss function is a general norm? Answering these problems could lead to a better understanding of Problem 1.

## Acknowledgements

P. Zhong is supported in part by NSF grants (CCF-1703925, CCF-1421161, CCF-1714818, CCF-1617955 and CCF-1740833), Simons Foundation (#491119), Google Research Award and a Google Ph.D. fellowship. R. Wang is supported in part by NSF grant IIS-1763562, Office of Naval Research (ONR) grants (N00014-18-1-2562, N00014-18-1-2861), and Nvidia NVAIL award. Part of this work was done while Z. Song, L. F. Yang, H. Zhang and P. Zhong were interns at IBM Research - Almaden and while Z. Song, R. Wang and H. Zhang were visiting the Simons Institute for the Theory of Computing. Z. Song and P. Zhong would like to thank Alexandr Andoni, Kenneth L. Clarkson, Yin Tat Lee, Eric Price, Clifford Stein and David P. Woodruff for insight discussions.

## Footnotes

[2] https://sublinear.info/index.php?title=Open_Problems:30

[3]Even for $\ell_p$ norms with $p < 2$, embeddings based on random projections will necessarily induce a distortion factor polynomial in $d$, as shown in [32].

[4]See, e.g., [24], for a survey on leverage scores.

[5]Alternatively, we can use the oblivious subspace embedding in [2] for this step. However, as we have discussed, the oblivious subspace embedding in [2] requires stronger assumptions on the function $G : \mathbb{R} \to \mathbb{R}_{\ge 0}$ than those in Assumption 1, which restricts the class of Orlicz norms to which our algorithm can be applied.

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
