[Supplementary Material]

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

# Supplementary Material for "Efficient Symmetric Norm Regression via Linear Sketching"

## A  Preliminaries

**Notations.**  For a matrix $A \in \mathbb{R}^{n \times d}$, we use $A_i$ to denote its $i$-th row, $A^i$ to denote its $i$-th column, $\|A\|_F$ to denote the Frobenius norm of $A$, and $\|A\|_2$ to denote the spectral norm of $A$. For any $n' \leq n$, we define $\xi^{(n')} \in \mathbb{R}^n$ to be a vector $\xi^{(n')} = \frac{1}{\sqrt{n'}}(1, 1, \ldots, 1, 0, 0, \ldots, 0)$.

**$\varepsilon$-nets.**  We use the standard upper bound on size of $\varepsilon$-nets.

**Definition 8.**  For a given set $\mathcal{S}$ and a norm $\|\cdot\|$, we say $\mathcal{N} \subseteq \mathcal{S}$ is a $\varepsilon$-net of $\mathcal{S}$ if for any $s \in \mathcal{S}$, there exists some $\overline{s} \in \mathcal{N}$ such that $\|s - \overline{s}\| \leq \varepsilon$.

**Lemma 13** ([33, II.E, 10]).  *Given a matrix $A \in \mathbb{R}^{n \times d}$ and a norm $\|\cdot\|$, let $\mathcal{S}$ be the unit $\|\cdot\|$-norm ball in the column space of $A$, i.e., $\mathcal{S} = \{Ax \mid \|Ax\| = 1\}$. For $\varepsilon \in (0, 1)$, there exists an $\varepsilon$-net $\mathcal{N}$ of $\mathcal{S}$ with size $|\mathcal{N}| \leq (1 + 1/\varepsilon)^d$.*

## B  Missing Proofs in Section 2

In this section, we give missing proofs in Section 2.

We first show that if a function $G$ satisfies Assumption 1, then $G$ has at least linear growth. We will use this fact in later proofs.

**Lemma 14.**  *Given a function $G$ that satisfies property $\mathcal{P}$, then for any $0 < x \leq y$, $y/x \leq G(y)/G(x)$.*

*Proof.* Due to the convexity of $G$ and $G(0) = 0$, for any $y > x > 0$, we have

$$G(x) \leq G(y)x/y + G(0)(1 - x/y) = G(y)x/y.$$

$\square$

### B.1  Proof of Lemma 3

*Proof.* The first condition is clear from the definition of $\|x\|_{G,w}$.

Now we prove the second condition. When $\|x + y\|_{G,w} = 0$, the triangle inequality clearly holds since $\|x\|_{G,w} \geq 0$ and $\|y\|_{G,w} \geq 0$. When $\|x\|_{G,w} = 0$ and $\|x + y\|_{G,w} \neq 0$, for any $\alpha > 0$, we have

$$\sum_{i=1}^{n} w_i G(|x_i + y_i|/\alpha) = \sum_{i|w_i>0} w_i G(|x_i + y_i|/\alpha) = \sum_{i|w_i>0} w_i G(|y_i|/\alpha) = \sum_{i=1}^{n} w_i G(|y_i|/\alpha),$$

which implies $\|x + y\|_{G,w} = \|y\|_{G,w}$. Similarly, the second condition also holds if $\|y\|_{G,w} = 0$ and $\|x + y\|_{G,w} \neq 0$. If $\|x + y\|_{G,w} \neq 0$, $\|x\|_{G,w} \neq 0$ and $\|y\|_{G,w} \neq 0$, by definition of $\|\cdot\|_{G,w}$, we have

$$\sum_{i=1}^{n} w_i G(x_i/\|x\|_{G,w}) = 1$$

and

$$\sum_{i=1}^{n} w_i G(y_i/\|y\|_{G,w}) = 1.$$

Thus,

$$\sum_{i=1}^{n} w_i G\left(\frac{x_i + y_i}{\|x\|_{G,w} + \|y\|_{G,w}}\right)$$

$$\leq \sum_{i=1}^{n} w_i G\left(\frac{|x_i| + |y_i|}{\|x\|_{G,w} + \|y\|_{G,w}}\right) \qquad (G \text{ is increasing})$$

$$\leq \sum_{i=1}^{n} w_i \left(\frac{\|x\|_{G,w}}{\|x\|_{G,w} + \|y\|_{G,w}} \cdot G(|x_i|/\|x\|_{G,w}) + \frac{\|y\|_{G,w}}{\|x\|_{G,w} + \|y\|_{G,w}} \cdot G(|y_i|/\|y\|_{G,w})\right)$$

$$\qquad\qquad (G \text{ is convex})$$

$$=1,$$

which implies $\|x + y\|_{G,w} \leq \|x\|_{G,w} + \|y\|_{G,w}$.

For the third condition, for any $a \in \mathbb{R}$ and $x \in \mathbb{R}^n$, if $\|x\|_{G,w} = 0$ then $\|ax\|_{G,w} = 0$. If $a = 0$, we have $\|ax\|_{G,w} = 0$. Otherwise, we have

$$\sum_{i=1}^{n} w_i G(x_i/\|x\|_{G,w}) = 1,$$

which implies

$$\sum_{i=1}^{n} w_i G\left(\frac{a \cdot x_i}{|a|\|x\|_{G,w}}\right) = 1,$$

and thus $\|ax\|_{G,w} = |a|\|x\|_{G,w}$. $\qquad\qquad\square$

### B.2 Proof of Lemma 4

*Proof.* Let $g \in \mathbb{R}^d$ be a vector whose entries are i.i.d. Gaussian random variables with zero mean and standard deviation $10^2$. We show that with probability at least $0.8$,

$$\sum_{i=1}^{n} G(\|U_i\|_2) \leq O\left(\sum_{i=1}^{n} G(\langle U_i, g\rangle)\right) \leq O\left(\max\{1, C_G\|Ug\|_G^2\}\right) \leq O(C_G d\kappa_G^2).$$

We divide our proofs into three parts.

**Part I.** We will show that with probability at least $0.9$,

$$\sum_{i=1}^{n} G(\|U_i\|_2) \leq O\left(\sum_{i=1}^{n} G(\langle U_i, g\rangle)\right).$$

For each $i \in [n]$, $\langle U_i, g\rangle$ has the same distribution as $10^2 \cdot \|U_i\|_2 \cdot \mathcal{N}(0, 1)$. For each $i \in [n]$, we let $B_i$ be the random variable such that

$$B_i = \begin{cases} 1 & |\langle U_i, g\rangle| \leq \|U_i\|_2 \\ 0 & \text{otherwise} \end{cases}.$$

By tail inequalities of standard Gaussian random variables, $\Pr[B_i = 1] \leq 0.01$. Thus,

$$\mathbf{E}\left[B_i \cdot G(\|U_i\|_2)\right] \leq 0.01 \cdot G(\|U_i\|_2),$$

which implies

$$\mathbf{E}\left[\sum_{i=1}^{n} B_i \cdot G(\|U_i\|_2)\right] \leq 0.01 \cdot \sum_{i=1}^{n} G(\|U_i\|_2),$$

By the monotonicity of $G$, since

$$G(\langle U_i, g\rangle) \geq (1 - B_i)G(\|U_i\|_2),$$

we have

$$\sum_{i=1}^{n} G(\langle U_i, g\rangle) \geq \sum_{i=1}^{n} (1 - B_i) G(\|U_i\|_2).$$

By Markov's inequality, with probability at least 0.9, we have

$$\sum_{i=1}^{n} B_i \cdot G(\|U_i\|_2) \leq 0.1 \sum_{i=1}^{n} G(\|U_i\|_2),$$

which implies

$$\sum_{i=1}^{n} G(\langle U_i, g\rangle) \geq 0.9 \sum_{i=1}^{n} G(\|U_i\|_2).$$

**Part II.** We will show that

$$\sum_{i=1}^{n} G(\langle U_i, g\rangle) \leq \max\{1, C_G \cdot \|Ug\|_G^2\}.$$

When $\|Ug\|_G \leq 1$, by monotonicity of $G$, we must have

$$\sum_{i=1}^{n} G(\langle U_i, g\rangle) \leq 1.$$

When $\|Ug\|_G \geq 1$, we have

$$\sum_{i=1}^{n} G(\langle U_i, g\rangle / \|Ug\|_G) = 1.$$

Since

$$G(\langle U_i, g\rangle) \leq G(\langle U_i, g\rangle / \|Ug\|_G) \cdot C_G \|Ug\|_G^2$$

and

$$\sum_{i=1}^{n} G(\langle U_i, g\rangle / \|Ug\|_G) = 1,$$

we must have

$$\sum_{i=1}^{n} G(\langle U_i, g\rangle) \leq \sum_{i=1}^{n} G(\langle U_i, g\rangle / \|Ug\|_G) \cdot C_G \cdot \|Ug\|_G^2 \leq C_G \cdot \|Ug\|_G^2.$$

**Part III.** We will show that $\|Ug\|_G^2 \leq O(C_G d\kappa_G^2)$. By definition of a well-conditioned basis and tail inequalities of Gaussian random variables, with probability at least 0.9, we have

$$\|Ug\|_G \leq \kappa_G \|g\|_2 \leq O(\kappa_G \sqrt{d}).$$

Thus, applying a union bound over three parts of the proof, we have with probability at least 0.8,

$$\sum_{i=1}^{n} G(\|U_i\|_2) \leq O(C_G d\kappa_G^2). \tag{9}$$

However, the condition in (9) is deterministic. Thus, the condition in (9) always holds. $\quad\square$

### B.3 Proof of Lemma 5

*Proof.* Notice that for any $x \in \mathbb{R}^d$,

$$\|AR^{-1}x\|_G \leq \|\Pi AR^{-1}x\|_2 = \kappa \|Qx\|_2 = \kappa \|x\|_2$$

and

$$\|AR^{-1}x\|_G \geq \frac{1}{\kappa} \|\Pi AR^{-1}x\|_2 = \|Qx\|_2 = \|x\|_2.$$

$\quad\square$

## B.4 Proof of Lemma 6

*Proof.* In Theorem 2.13 of [34], it has been shown how to calculate $\{l_i\}_{i=1}^n$ such that $l_i = \Theta(\|(AR^{-1})_i\|_2)$ in $\widetilde{O}(\mathrm{nnz}(A) + \mathrm{poly}(d))$ time with probability at least 0.99. We simply take $u_i = G(l_i)$. By Lemma 14 and the growth condition of $G$, we must have $u_i = \Theta(G(\|(AR^{-1})_i\|_2))$. $\square$

## B.5 Proof of Lemma 7

*Proof.* By homogeneity, we only need to prove that with probability $1 - \delta$, for all $x$ which satisfies $\|Ux\|_G = 1$,
$$(1-\varepsilon)\|Ux\|_G \leq \|Ux\|_{G,w} \leq (1+\varepsilon)\|Ux\|_G.$$

We first prove that for any fixed $x \in \mathbb{R}^d$ such that $\|Ux\|_G = 1$, with probability $1 - \delta(1 + 4/\varepsilon)^{-d}$,
$$(1-\varepsilon/4)\|Ux\|_G \leq \|Ux\|_{G,w} \leq (1+\varepsilon/4)\|Ux\|_G.$$

Let $x \in \mathbb{R}^d$ that satisfies $\|Ux\|_G = 1$ and $y = Ux$. Let $Z_i$ be a random variable which denotes the value of $w_i G(y_i)$ and $Z = \sum_{i=1}^n Z_i$.

We will first show that if $Z \in [1 - \varepsilon/4, 1 + \varepsilon/4]$, then $\|y\|_{G,w} \in [1 - \varepsilon/4, 1 + \varepsilon/4]$. There are three cases:

1. If $\|y\|_{G,w} = 1$, then $\|y\|_{G,w}$ is already in $[1 - \varepsilon/4, 1 + \varepsilon/4]$.

2. If $\|y\|_{G,w} > 1$, then by Lemma 14, we have
$$\sum_{i=1}^n w_i G(y_i) \geq \sum_{i=1}^n w_i \|y\|_{G,w} \cdot G(y_i/\|y\|_{G,w}).$$
   Since
$$\sum_{i=1}^n w_i \cdot G(y_i/\|y\|_{G,w}) = 1,$$
   we must have
$$\|y\|_{G,w} \leq \sum_{i=1}^n w_i G(y_i) = Z \leq 1 + \varepsilon/4.$$

3. If $\|y\|_{G,w} < 1$, then by Lemma 14, we have
$$1 = \sum_{i=1}^n w_i G(y_i/\|y\|_{G,w}) \geq 1/\|y\|_{G,w} \cdot \sum_{i=1}^n w_i G(y_i),$$
   which implies
$$\|y\|_{G,w} \geq \sum_{i=1}^n w_i G(y_i) = Z \geq 1 - \varepsilon/4.$$

Thus, it suffices to prove that
$$\Pr\left[Z \in [1 - \varepsilon/4, 1 + \varepsilon/4]\right] \geq 1 - \delta(1 + 4/\varepsilon)^{-d}.$$
Consider the expectation of $Z$, we have
$$\mathbf{E}[Z] = \sum_{i=1}^n \mathbf{E}[Z_i] = \sum_{i=1}^n \mathbf{E}[w_i] \cdot G((Ux)_i) = \sum_{i=1}^n G((Ux)_i) = 1,$$
where the last equality follows since $\|Ux\|_G = 1$.

Notice that $|Z_i - \mathbf{E}(Z_i)|$ is always upper bounded by
$$w_i G(y_i) = w_i G((Ux)_i) \leq w_i G(\|U_i\|_2 \cdot \|x\|_2) \leq w_i G(\|U_i\|_2)$$
$$\leq G(\|U_i\|_2)/p_i \leq \frac{\varepsilon^2}{C(\log(1/\delta) + d\log(1/\varepsilon))},$$

where the first inequality follows from Cauchy-Schwarz inequality, the second inequality follows from the definition of well-conditioned basis in Definition 2 and monotonicity of $G$, the third inequality follows from definition of $w_i$ and the last inequality follows from the choice of $p_i$.

Consider the variance of $Z$, we have:

$$\mathrm{Var}(Z) = \sum_{i|p_i<1} \mathrm{Var}(Z_i) \leq \sum_{i|p_i<1} \mathbf{E}(Z_i^2) = \sum_{i|p_i<1} \left(G((Ux)_i)\right)^2/p_i$$

$$\leq \left(\sum_{i|p_i<1} G((Ux)_i)\right) \cdot \max_{i|p_i<1} G((Ux)_i)/p_i \leq \frac{\varepsilon^2}{C\left(\log(1/\delta) + d\log(1/\varepsilon)\right)},$$

where the second inequality follows from Hölder's inequality and the last inequality follows from the upper bound of $G((Ux)_i)/p_i$ and $\|Ux\|_G = 1$.

Thus, by Bernstein inequality, we have:

$$\Pr\left(|Z - 1| > \varepsilon/4\right) \leq (1 + 4/\varepsilon)^{-d}\delta.$$

Thus, for a fixed $x$, with probability at least $1 - (1 + 4/\varepsilon)^{-d}\delta$, we have

$$(1 - \varepsilon/4)\|Ux\|_G \leq \|Ux\|_{G,w} \leq (1 + \varepsilon/4)\|Ux\|_G.$$

Let $\mathcal{S}$ be the unit $\|\cdot\|_G$-norm ball in the column space of $U$, i.e., $\mathcal{S} = \{Ux \mid \|Ux\|_G = 1\}$. According to Lemma 13, there exists an $\varepsilon/4$-net $\mathcal{N}$ of $\mathcal{S}$ with $|\mathcal{N}| \leq (1 + 4/\varepsilon)^d$. We use $\mathcal{E}$ to denote the event that for all $y \in \mathcal{N}$, $\|y\|_{G,w} \in [1 - \varepsilon/4, 1 + \varepsilon/4]$. By taking union bound over all vectors in $\mathcal{N}$, we have $\Pr[\mathcal{E}] \geq 1 - \delta$.

Conditioned on $\mathcal{E}$, now we show that for all $y \in \mathcal{S}$, $\|y\|_{G,w} \in [1 - \varepsilon, 1 + \varepsilon]$. Consider a fixed vector $y \in \mathcal{S}$, since $\mathcal{N}$ is an $\varepsilon/4$-net of $\mathcal{S}$, we can choose a vector $u^{(1)} \in \mathcal{N}$ such that

$$\|y - u^{(1)}\|_G \leq \varepsilon/4.$$

Thus, we have that

$$\|y\|_{G,w} \leq \|u^{(1)}\|_{G,w} + \|y - u^{(1)}\|_{G,w} \leq (1 + \varepsilon/4) + \|y - u^{(1)}\|_{G,w}.$$

Let $\alpha^{(1)} = 1/\|y - u^{(1)}\|_G$. Then we have $\alpha^{(1)}(y - u^{(1)}) \in \mathcal{S}$. Thus, there exist $u^{(2)} \in \mathcal{N}$ such that

$$\|u^{(2)} - \alpha^{(1)}(y - u^{(1)})\|_G \leq \varepsilon/4.$$

It implies that

$$\|(y - u^{(1)}) - u^{(2)}/\alpha^{(1)}\|_G \leq \varepsilon/(4\alpha^{(1)}) \leq (\varepsilon/4)^2.$$

Thus,

$$\|y - u^{(1)}\|_{G,w} \leq \|u^{(2)}\|_{G,w}/\alpha^{(1)} + \|y - u^{(1)} - u^{(2)}/\alpha^{(1)}\|_{G,w} \leq (1 + \varepsilon/4)\varepsilon/4 + \|y - u^{(1)} - u^{(2)}/\alpha^{(1)}\|_{G,w}.$$

Let $\alpha^{(2)} = 1/\|y - u^{(1)} - u^{(2)}/\alpha^{(1)}\|_G$. Then we can repeat the above argument and get

$$\|y\|_{G,w} \leq (1 + \varepsilon/4) + (1 + \varepsilon/4)\varepsilon/4 + (1 + \varepsilon/4)(\varepsilon/4)^2 + \dots$$
$$= (1 + \varepsilon/4)/(1 - \varepsilon/4) \leq 1 + \varepsilon.$$

By applying the above upper bound on $\|\alpha^{(1)}(u^{(1)} - y)\|_{G,w}$, we can get

$$\|y\|_{G,w} \geq \|u^{(1)}\|_{G,w} - \|u^{(1)} - y\|_{G,w}$$
$$\geq (1 - \varepsilon/4) - \|u^{(1)} - y\|_{G,w}$$
$$\geq (1 - \varepsilon/4) - \frac{1 + \varepsilon}{\alpha^{(1)}}$$
$$\geq 1 - \varepsilon/2 - \varepsilon^2/4$$
$$\geq 1 - \varepsilon.$$

Thus, conditioned on $\mathcal{E}$, which holds with probability $1 - \delta$, we have $\|y\|_{G,w} \in [1 - \varepsilon, 1 + \varepsilon]$ for all $y = Ux$ with $\|y\|_G = 1$. $\qquad\square$

## B.6 Proof of Theorem 8

*Proof.* We first analyze the running time of the algorithm. In Step 1, we calculate $\Pi\overline{A}$ and invoke QR-decomposition on $\Pi\overline{A}$. In Step 2, we apply the algorithm in Lemma 6, which runs in $\widetilde{O}(\mathrm{nnz}(A) + \mathrm{poly}(d))$ time. Obtaining the weight vector $w \in \mathbb{R}^n$ in Step 3 requires $O(n)$ time.

Since for all $x \in \mathbb{R}^d$,
$$\|\overline{A}x\|_G \leq \|\Pi\overline{A}x\|_2 \leq \kappa\|\overline{A}x\|_G.$$
we have

$$
\begin{aligned}
\mathbf{E}[\|w\|_0] = \sum_{i=1}^n p_i = \sum_{i=1}^n & O\left(d\log(1/\varepsilon)/\varepsilon^2 \cdot G\left(\|(\overline{A}R^{-1})_i\|_2\right)\right) \\
& \leq O\left(d\log(1/\varepsilon)/\varepsilon^2 \cdot C_G d\left(\kappa_G(\overline{A}R^{-1})\right)^2\right) \quad\text{(Lemma 4)} \\
& \leq O\left(C_G d^2 \kappa^2 \log(1/\varepsilon)/\varepsilon^2\right). \quad\text{(Lemma 5)}
\end{aligned}
$$

By Markov's inequality, with constant probability we have $\|w\|_0 \leq O\left(C_G d^2 \kappa^2 \log(1/\varepsilon)/\varepsilon^2\right)$. Moreover, in order to solve $\min_x \|Ax - b\|_{G,w}$, we can ignore all rows of $A$ with zero weights, and thus there are at most $O\left(C_G d^2 \kappa^2 \log(1/\varepsilon)/\varepsilon^2\right)$ remaining rows in $A$. Furthermore, as we show in Lemma 3, $\|\cdot\|_{G,w}$ is a seminorm, which implies we can solve $\min_x \|Ax - b\|_{G,w}$ in $\mathrm{poly}(C_G d\kappa/\varepsilon)$ time, by simply solve a convex program with size $O\left(C_G d^2 \kappa^2 \log(1/\varepsilon)/\varepsilon^2\right)$.

Now we prove the correctness of the algorithm. The algorithm in Lemma 6 succeeds with constant probability. By Lemma 7, with constant probability, simultaneously for all $x \in \mathbb{R}^{d+1}$,
$$(1 - \varepsilon/3)\|\overline{A}R^{-1}x\|_G \leq \|\overline{A}R^{-1}x\|_{G,w} \leq (1 + \varepsilon/3)\|\overline{A}R^{-1}x\|_G.$$
Equivalently, with constant probability, simultaneously for all $x \in \mathbb{R}^{d+1}$,
$$(1 - \varepsilon/3)\|\overline{A}x\|_G \leq \|\overline{A}x\|_{G,w} \leq (1 + \varepsilon/3)\|\overline{A}x\|_G.$$

Since $x^* = \mathrm{argmin}_x \|Ax - b\|_{G,w}$, for all $x \in \mathbb{R}^d$, we have
$$
\begin{aligned}
\|Ax^* - b\|_G &\leq 1/(1-\varepsilon/3)\|Ax^* - b\|_{G,w} \leq 1/(1-\varepsilon/3)\|Ax - b\|_{G,w} \\
&\leq (1+\varepsilon/3)/(1-\varepsilon/3)\|Ax - b\|_G \leq (1+\varepsilon)\|Ax - b\|_G
\end{aligned}
$$
for sufficiently small $\varepsilon$. Thus, $x^*$ is a $(1 + \varepsilon)$-approximate solution to $\min_x \|Ax - b\|_G$.

Note that the failure probability of the algorithm can be reduced to an arbitrarily small constant by independent repetitions and taking the best solution found among all repetitions. □

## C Missing Proofs in Section 3

In this section, we give missing proofs in Section 3.

Without loss of generality, throughout this section, for the symmetric norm $\|\cdot\|_\ell$ under consideration, we assume $\|\xi^{(1)}\|_\ell = 1$.

### C.1 Background

#### C.1.1 Known $\ell_2$ Oblivious Subspace Embeddings

In this section, we recall some known $\ell_2$ subspace embeddings.

**Definition 9.** We say $S \in \mathbb{R}^{t \times n}$ is an $\ell_2$ subspace embedding for the column space of $A \in \mathbb{R}^{n \times d}$, if for all $x \in \mathbb{R}^d$,
$$(1 - \varepsilon)\|Ax\|_2 \leq \|SAx\|_2 \leq (1 + \varepsilon)\|Ax\|_2.$$

**Definition 10.** A CountSketch embedding is defined to be $\Pi = \Phi D \in \mathbb{R}^{m \times n}$ with $m = \Theta(d^2/\varepsilon^2)$, where $D$ is an $n \times n$ random diagonal matrix with each diagonal entry independently chosen to be $+1$ or $-1$ with equal probability, and $\Phi \in \{0,1\}^{m \times n}$ is an $m \times n$ binary matrix with $\Phi_{h(i),i} = 1$ and all remaining entries being 0, where $h : [n] \to [m]$ is a random map such that for each $i \in [n]$, $h(i) = j$ with probability $1/m$ for each $j \in [m]$.

**Theorem 15** ([12]). *For a given matrix $A \in \mathbb{R}^{n \times d}$ and $\varepsilon \in (0, 1/2)$. Let $\Pi \in \mathbb{R}^{\Theta(d^2/\varepsilon^2) \times n}$ be a* CountSketch *embedding. With probability at least $0.9999$, $\Pi$ is an $\ell_2$ subspace embedding for the column space of $A$. Furthermore, $\Pi A$ can be computed in $O(\mathrm{nnz}(A))$ time.*

**Definition 11.** A Gaussian embedding $S$ is defined to be $\frac{1}{\sqrt{m}} \cdot G \in \mathbb{R}^{m \times n}$ with $m = \Theta(d/\varepsilon^2)$, where each entry of $G \in \mathbb{R}^{m \times n}$ is chosen independently from the standard Gaussian distribution.

**Theorem 16** ([34]). *For a given matrix $A \in \mathbb{R}^{n \times d}$ and $\varepsilon \in (0, 1/2)$. Let $S \in \mathbb{R}^{\Theta(d/\varepsilon^2) \times n}$ be a Gaussian embedding. With probability at least $0.9999$, $S$ is an $\ell_2$ subspace embedding for the column space of $A$.*

**Definition 12.** A composition of Gaussian embedding and CountSketch embedding is defined to be $S' = S\Pi$, where $\Pi \in \mathbb{R}^{\Theta(d^2/\varepsilon^2) \times n}$ is a CountSketch embedding and $S \in \mathbb{R}^{\Theta(d/\varepsilon^2) \times \Theta(d^2/\varepsilon^2)}$ is a Gaussian embedding.

The following corollary directly follows from the above two theorems.

**Corollary 17.** *For a given matrix $A \in \mathbb{R}^{n \times d}$ and $\varepsilon \in (0, 1/2)$. Let $S' \in \mathbb{R}^{\Theta(d/\varepsilon^2) \times n}$ be a composition of Gaussian embedding and* CountSketch *embedding. With probability at least $0.9998$, $S'$ is an $\ell_2$ subspace embedding for the column space of $A$. Furthermore, $S'A$ can be computed in $O(\mathrm{nnz}(A) + d^4/\varepsilon^4)$ time.*

We remark that all $\ell_2$ subspace embeddings introduced in this section are *oblivious*, meaning that the distribution of the embedding matrix does not depend on the matrix $A$.

### C.1.2 Properties of Symmetric Norms

**General Properties.** We first introduce several general properties of symmetric norms.

**Lemma 18** (Lemma 2.1 in [7]). *For any symmetric norm $\| \cdot \|_\ell$ and $x, y \in \mathbb{R}^n$ such that for all $i \in [n]$ we have $|x_i| \leq |y_i|$, then $\|x\|_\ell \leq \|y\|_\ell$.*

**Lemma 19** (Fact 2.2 in [7]). *Suppose $\|\xi^{(1)}\|_\ell = 1$, for any vector $x \in \mathbb{R}^n$,*

$$\|x\|_\infty \leq \|x\|_\ell \leq \|x\|_1.$$

**Lemma 20** (Lemma 3.12 in [7]). *Let $\| \cdot \|_\ell$ be a symmetric norm. Then*

$$\Omega(M_\ell/\sqrt{\log n}) \leq \|\xi^{(n)}\|_\ell \leq O(M_\ell),$$

*where $M_\ell$ is as defined in Definition 6.*

**Modulus of Approximation.** We need the following quantity of a symmetric norm.

**Definition 13.** The *maximum modulus of approximation* of a symmetric norm $\| \cdot \|_\ell$ is defined as

$$\mathrm{mma}(\ell, r) = \max_{1 \leq a \leq b \leq ar \leq n} \frac{M_{\ell^{(ar)}}}{M_{\ell^{(b)}}},$$

where $\| \cdot \|_{\ell^{(k)}}$ is a norm on $\mathbb{R}^k$ which is defined to be

$$\|(x_1, x_2, \ldots, x_k)\|_{\ell^{(k)}} = \|(x_1, x_2, \ldots, x_k, 0, \ldots, 0)\|_\ell,$$

and $M_{\ell^{(k)}}$ is as defined in Definition 6.

Intuitively, $\mathrm{mma}(\ell, r)$ characterizes how well the original symmetric norm can be approximated by a lower dimensional induced norm. We show in the following lemma that $\mathrm{mma}(\ell, r) \leq O(\sqrt{r \log n})$ for any symmetric norm.

**Lemma 21.** *For any symmetric norm $\| \cdot \|_\ell$ and $r \in [n]$, $\mathrm{mma}(\ell, r) \leq O(\sqrt{r \log n})$.*

*Proof.* By Lemma 20, for any $i \in [n]$, $\Omega(M_{\ell^{(i)}}/\sqrt{\log n}) \leq \|\xi^{(i)}\|_\ell \leq O(M_{\ell^{(i)}})$. Let $ar = c_1 b + c_2$, where $c_1, c_2$ are non-negative integers with $c_1 \leq ar/b$ and $c_2 \leq b$. Observe that we can rewrite $\xi^{(ar)}$ as

$$\xi^{(ar)} = \left[ \frac{\sqrt{b}}{\sqrt{ar}} \cdot \left( \underbrace{\xi^{(b)}, \xi^{(b)}, \xi^{(b)}, \ldots, \xi^{(b)}}_{c_1 \text{ times}} \right), \frac{\sqrt{c_2}}{\sqrt{ar}} \xi^{(c_2)}, 0, \ldots, 0 \right].$$

Therefore, by triangle inequality, we have

$$\|\xi^{(ar)}\|_\ell \leq \frac{\sqrt{b}}{\sqrt{ar}} \cdot c_1 \cdot \|\xi^{(b)}\|_\ell + \frac{\sqrt{c_2}}{\sqrt{ar}} \cdot \|\xi^{(c_2)}\|_\ell$$

$$\leq \sqrt{\frac{b}{ar}} \cdot \frac{ar}{b} \cdot \|\xi^{(b)}\|_\ell + \frac{\sqrt{b}}{\sqrt{ar}} \cdot \|\xi^{(b)}\|_\ell \qquad (c_1 \leq ar/b \text{ and } c_2 \leq b)$$

$$\leq \sqrt{r} \cdot \|\xi^{(b)}\|_\ell + \|\xi^{(b)}\|_\ell \qquad\qquad (a \leq b \leq ar)$$

$$\leq 2\sqrt{r} \cdot \|\xi^{(b)}\|_\ell. \qquad\qquad (r \geq 1)$$

Now we apply Lemma 20 on both sides, which implies

$$\frac{M_{\ell^{(ar)}}}{\sqrt{\log n}} \leq O(\sqrt{r} \cdot M_{\ell^{(b)}})$$

as desired. $\qquad\qquad\square$

**Properties of SymSketch.** Now we introduce several properties of SymSketch.

The following lemma shows that for a data matrix $A \in \mathbb{R}^d$, calculating $SA$ requires $\widetilde{O}(\mathrm{nnz}(A)) + \mathrm{poly}(d))$ time for a SymSketch $S$.

**Lemma 22.** *For a given matrix $A \in \mathbb{R}^{n \times d}$, let $S \in \mathbb{R}^{O(d) \times n}$ be a SymSketch as in Definition 5. $SA$ can be computed in $O(\mathrm{nnz}(A)) + \mathrm{poly}(d)$ time in expectation, and in $O(\mathrm{nnz}(A) \log n) + \mathrm{poly}(d)$ time in the worst case.*

*Proof.* Since $S$ is a SymSketch, $S = \Pi\widetilde{D} = \Pi \cdot \begin{bmatrix} w_0 D_0 \\ w_1 D_1 \\ \vdots \\ w_t D_t \end{bmatrix}$, where $\Pi \in \mathbb{R}^{O(d) \times O(n \log n)}$.

Since $D_i$ is a diagonal matrix, $\mathrm{nnz}(D_i A) \leq \mathrm{nnz}(A)$, and thus $\mathrm{nnz}(\widetilde{D}A) \leq (t+1) \cdot \mathrm{nnz}(A) = O(\mathrm{nnz}(A) \log n)$, which implies $\widetilde{D}A$ can be computed in $(t+1) \cdot \mathrm{nnz}(A) = O(\mathrm{nnz}(A) \log n)$ time.

On the other hand, the expected number of nonzero entries of $D_i A$ is $2^{-i} \mathrm{nnz}(A)$. Thus, $\widetilde{D}A$ has $O(\mathrm{nnz}(A))$ nonzero entries in expectation, which implies $\widetilde{D}A$ can be computed in $O(\mathrm{nnz}(A))$ time.

Finally, notice that $\Pi$ is a composition of Gaussian embedding and CountSketch embedding, which implies $\Pi\widetilde{D}A$ can be computed in $\mathrm{nnz}(\widetilde{D}A) + \mathrm{poly}(d)$ time. $\qquad\square$

The following lemma shows that with constant probability, for all $x \in \mathbb{R}^n$, $\|Sx\|_2 \leq \mathrm{poly}(n)\|x\|_2$.

**Lemma 23.** *Let $S \in \mathbb{R}^{O(d) \times n}$ be a SymSketch as defined in Definition 5, then with probability at least $0.9999$ $\|S\|_2 \leq \mathrm{poly}(n)$.*

*Proof.* Notice that $S = \Pi\widetilde{D}$, since $\|S\|_2 \leq \|\Pi\|_2 \cdot \|\widetilde{D}\|_2$, it suffices to bound $\|\Pi\|_2$ and $\|\widetilde{D}\|_2$. Since $\Pi$ is a composition of Gaussian embedding and CountSketch embedding (Definition 12), with probability at least $0.9999$, $\|\Pi\|_2 \leq \|\Pi\|_F \leq \mathrm{poly}(n)$. Now consider $\widetilde{D} = \begin{bmatrix} w_0 D_0 \\ w_1 D_1 \\ \vdots \\ w_t D_t \end{bmatrix}$. By Lemma 19, for all $j \in [t]$, $w_j \leq \mathrm{poly}(n)$. Furthermore, $\|D_j\|_2 \leq 1$ and $t = \Theta(\log n)$, which implies $\|\widetilde{D}\|_2 \leq \mathrm{poly}(n)$. $\qquad\square$

Throughout this whole section we assume that for any non-zero vector $x \in \mathbb{R}^n$, we have $1 \leq |x_j| \leq \mathrm{poly}(n)$ for all $j \in [n]$. Notice that this assumption is without loss of generality, as shown in the following lemma.

**Lemma 24.** *For any non-zero vector $x \in \mathbb{R}^n$, let $\overline{x} \in \mathbb{R}^n$ be a vector with $\overline{x} = \frac{\mathrm{poly}(n) \cdot x}{\|x\|_\infty}$, and $x' \in \mathbb{R}^n$ where*

$$x_i' = \begin{cases} \overline{x}_i & \text{if } \overline{x}_i \geq 1 \\ 0 & \text{otherwise} \end{cases}.$$

*For a symmetric norm $\| \cdot \|_\ell$, suppose $\|S\|_2 \leq \mathrm{poly}(n)$ and*

$$\alpha \|x'\|_\ell \leq \|Sx'\|_2 \leq \beta \|x'\|_\ell$$

*for some $\alpha, \beta \in [1/\mathrm{poly}(n), \mathrm{poly}(n)]$, then*

$$\Omega(\alpha) \|x\|_\ell \leq \|Sx\|_2 \leq O(\beta) \|x\|_\ell.$$

*Proof.* By triangle inequality and Lemma 18, we have

$$\|\overline{x}\|_\ell - n \leq \|x'\|_\ell \leq \|\overline{x}\|_\ell.$$

By Lemma 19,

$$\|x'\|_\ell \geq \|x'\|_\infty = \|\overline{x}\|_\infty = \mathrm{poly}(n),$$

we have

$$(1 - 1/\mathrm{poly}(n)) \|\overline{x}\|_\ell \leq \|x'\|_\ell \leq \|\overline{x}\|_\ell.$$

Notice that $\|S\overline{x}\|_2 = \|Sx' + S(\overline{x} - x')\|_2$. By triangle inequality we have

$$\|Sx'\|_2 - \|S\|_2 \|\overline{x} - x'\|_2 \leq \|S\overline{x}\|_2 \leq \|Sx'\|_2 + \|S\|_2 \|\overline{x} - x'\|_2.$$

By the given conditions, we have

$$(1 - 1/\mathrm{poly}(n)) \|Sx'\|_2 \leq \|S\overline{x}\|_2 \leq (1 + 1/\mathrm{poly}(n)) \|Sx'\|_2,$$

which implies

$$\Omega(\alpha) \|\overline{x}\|_\ell \leq \|S\overline{x}\|_2 \leq O(\beta) \|\overline{x}\|_\ell.$$

Since $\overline{x} = \frac{\mathrm{poly}(n) \cdot x}{\|x\|_\infty}$, we have

$$\Omega(\alpha) \|x\|_\ell \leq \|Sx\|_2 \leq O(\beta) \|x\|_\ell.$$

$\square$

By Lemma 24, we can focus on those non-zero vectors $x \in \mathbb{R}^n$ such that $1 \leq |x_j| \leq \mathrm{poly}(n)$ for all $j \in [n]$.

**Definition 14.** For a given vector $x \in \mathbb{R}^n$, suppose for all $j \in [n]$,

$$1 \leq |x_j| \leq \mathrm{poly}(n).$$

Let $g = \Theta(\log n)$. For each $i \in \{0, 1, \ldots, g\}$, we define

$$L_i(x) = \{j \mid 2^i \leq |x_j| < 2^{i+1}\}.$$

For each $i \in \{0, 1, \ldots, g\}$, we define $V_i(x) \in \mathbb{R}^n$ to be the vector

$$V_i(x) = (\underbrace{2^i, 2^i, \ldots, 2^i}_{|L_i(x)|}, 0, \ldots, 0).$$

For each $i \in \{0, 1, \ldots, g\}$, we say a level $i$ to be *contributing* if

$$\|V_i(x)\|_\ell \geq \Omega(1/g) \cdot \|x\|_\ell.$$

**Lemma 25.** *Let $g = \Theta(\log n)$. For a given vector $x \in \mathbb{R}^n$ such that for all $j \in [n], 1 \leq |x_j| \leq 2^g$, there exists at least one level $i \in \{0, 1, \ldots, g\}$ which is contributing.*

*Proof.* If none of $i \in \{0, 1, \ldots, g\}$ is contributing, then $\|x\|_\ell \leq \sum_{i=0}^{g} \|V_i(x)\|_\ell \leq 1/(2g) \cdot \sum_{i=0}^{g} \|x\|_\ell \leq \frac{1}{2} \|x\|_\ell$, which leads to a contradiction. $\square$

## C.2 Proof of Lemma 9

*Proof.* Consider a fixed $n' \in [n]$. By Lemma 20, we have

$$M_{\ell_a^{(n')}} = \Omega\left(\|\xi^{(n')}\|_{\ell_a^{(n')}}\right) = \Omega\left(1 + c\sqrt{n'}\right)$$

and

$$M_{\ell_b^{(n')}} = \Omega\left(\|\xi^{(n')}\|_{\ell_b^{(n')}}\right) = \Omega\left(\max\left(1, c\sqrt{n'}\right)\right).$$

It is also straightforward to verify that

$$\max_{x \in \mathbb{S}^{n'-1}} \|x\|_{\ell_a} = 1 + c\sqrt{n'}$$

and

$$\max_{x \in \mathbb{S}^{n'-1}} \|x\|_{\ell_b} = \max\left(1, c\sqrt{n'}\right).$$

Taking the ratio between $\max_{x \in \mathbb{S}^{n'-1}} \|x\|_\ell$ and $M_{\ell^{(n')}}$ for $\ell \in \{\ell_a, \ell_b\}$, we complete the proof. $\square$

## C.3 Proof of Lemma 10

*Proof.* Let $\overline{G}(x) = \sqrt{x} \cdot G^{-1}(1/x)$, where $G^{-1}(1/x)$ is the unique value in $[0, \infty)$ such that $G(G^{-1}(1/x)) = 1/x$. We first show that $\overline{G}(x)$ is an approximately decreasing function for $x \in (0, \infty)$. Let $m, n$ be two real numbers with $0 < m \le n$. We have $1/n \le 1/m$, which implies $0 < G^{-1}(1/n) \le G^{-1}(1/m)$ by monotonicity of $G$. By the third condition in Assumption 1, we have

$$\frac{G(G^{-1}(1/m))}{G(G^{-1}(1/n))} \le C_G \cdot \left(\frac{G^{-1}(1/m)}{G^{-1}(1/n)}\right)^2,$$

which implies

$$\sqrt{n} \cdot G^{-1}(1/n) \le \sqrt{C_G} \cdot \sqrt{m} \cdot G^{-1}(1/m).$$

Hence $\overline{G}(n) \le \sqrt{C_G} \cdot \overline{G}(m)$.

We are now ready to prove the lemma. Recall that for the Orlicz norm $\|\cdot\|_\ell = \|\cdot\|_G$, we have

$$\mathrm{mmc}(\ell) = \max_{n' \in [n]} \mathrm{mc}(\ell^{(n')}) = \max_{n' \in [n]} \frac{\max_{x \in \mathbb{S}^{n'-1}} \|x\|_{\ell^{(n')}}}{M_{\ell^{(n')}}}.$$

By Lemma 20, we have,

$$\Omega(1) \cdot \|\xi^{(n')}\|_{\ell^{(n')}} \le M_{\ell^{(n')}} \le O(\sqrt{\log n}) \cdot \|\xi^{(n')}\|_{\ell^{(n')}}.$$

Thus $\|\xi^{(n')}\|_{\ell^{(n')}}$ provides an approximation to $M_{\ell^{(n')}}$. By definition of $\|\cdot\|_G$, we have

$$\|\xi^{(n')}\|_{\ell^{(n')}} = \frac{1}{\sqrt{n'} \cdot G^{-1}(1/n')} = \frac{1}{\overline{G}(n')}.$$

Hence

$$\Omega(1) \le M_{\ell^{(n')}} \cdot \overline{G}(n') \le O(\sqrt{\log n}).$$

Next, we compute $\max_{x \in \mathbb{S}^{n'-1}} \|x\|_{\ell^{(n')}}$. For an arbitrary unit vector $x \in \mathbb{S}^{n'-1}$, we denote

$$B_j = \{i \in [n] : |x_i| \in [1/2^j, 1/2^{j-1})\}$$

and $b_j = |B_j|$. For each $j$, let $x^{B_j} \in \mathbb{R}^n$ be the vector such that

$$x_i^{B_j} = \begin{cases} x_i & \text{if } j \in B_j \\ 0 & \text{otherwise} \end{cases}.$$

Note that non-zero coordinates in $x^{B_j}$ have magnitude close to each other (within a factor of 2), we thus have

$$\|x^{B_j}\|_\ell = \|x^{B_j}\|_2 \cdot \left\|\frac{x^{B_j}}{\|x^{B_j}\|_2}\right\|_\ell \le \frac{\sqrt{b_j}}{2^{j-1}} \cdot \left\|\frac{x^{B_j}}{\|x^{B_j}\|_2}\right\|_\ell \le \frac{\sqrt{b_j}}{2^{j-2}} \cdot \|\xi^{(b_j)}\|_{\ell^{(b_j)}} = \frac{\sqrt{b_j}}{2^{j-2}} \cdot \frac{1}{\overline{G}(b_j)}.$$

Similarly,

$$\|x^{B_j}\|_\ell \geq \frac{\sqrt{b_j}}{2^{j+2}} \cdot \|\xi^{(b_j)}\|_{\ell^{(b_j)}} = \frac{\sqrt{b_j}}{2^{j+2}} \cdot \frac{1}{\overline{G}(b_j)} \geq \frac{\sqrt{b_j}}{2^{j+2}} \cdot \frac{1}{\sqrt{C_G} \cdot \overline{G}(1)}.$$

We claim there exists an constant $c > 0$ such that

$$\sum_{\substack{j > c \log n \\ b_j > 0}} \|x^{B_j}\|_\ell \leq \sum_{j' \leq c \log n} \|x^{B^{j'}}\|_\ell.$$

To show this, by Lemma 14, for any $b \geq 1$,

$$b = \frac{G(G^{-1}(1))}{G(G^{-1}(1/b))} \geq \frac{G^{-1}(1)}{G^{-1}(1/b)},$$

which implies

$$G^{-1}(1/b) \geq \frac{G^{-1}(1)}{b}.$$

Next, since $\|x\|_2 = 1$, there exists an $0 \leq \widetilde{j} \leq 4 \log n$ such that $b_{\widetilde{j}} \geq 1$. Therefore,

$$\|x^{B_{\widetilde{j}}}\|_\ell \geq \frac{1}{2^{\widetilde{j}+2}} \cdot \frac{1}{\sqrt{C_G} \cdot \overline{G}(1)}.$$

Thus, we have

$$\sum_{\substack{j > c \log n \\ b_j > 0}} \|x^{B_j}\|_\ell = \sum_{\substack{j > c \log n \\ b_j > 0}} \|x^{B_j}\|_2 \cdot \left\| \frac{x^{B_j}}{\|x^{B_j}\|_2} \right\|_\ell \leq \sum_{\substack{j > c \log n \\ b_j > 0}} \frac{\sqrt{n}}{2^{j-2}} \cdot \|\xi^{(b_j)}\|_{\ell^{(b_j)}}$$

$$\leq \sum_{\substack{j > c \log n \\ b_j > 0}} \frac{\sqrt{n}}{2^{j-2}} \cdot \frac{1}{\sqrt{b_j} G^{-1}(1/b_j)} \leq n \cdot \frac{\sqrt{n}}{2^{c \log n - 2}} \cdot \frac{b_j}{\sqrt{b_j} G^{-1}(1)}$$

$$\leq n \cdot 2^{\widetilde{j}+2} \cdot \frac{\sqrt{n}}{2^{c \log n - 2}} \cdot \sqrt{C_G n} \cdot \|x^{B_{\widetilde{j}}}\|_\ell \leq \|x^{B_{\widetilde{j}}}\|_\ell \leq \sum_{j' \leq c \log n} \|x^{B^{j'}}\|_\ell$$

for some sufficiently large constant $c$.

Let

$$j^* = \operatorname{argmax}_{j \leq c \log n} \|x^{B_j}\|_\ell,$$

we have

$$\|x^{B_{j^*}}\|_\ell \leq \|x\|_\ell \leq \sum_{j \leq c \log n} \|x^{B_j}\|_\ell + \sum_{\substack{j > c \log n \\ b_j > 0}} \|x^{B_j}\|_\ell \leq 2 \sum_{j \leq c \log n} \|x^{B_j}\|_\ell \leq O(\log n) \cdot \|x^{B_{j^*}}\|_\ell.$$

Thus,

$$\max_{x \in \mathbb{S}^{n'-1}} \|x\|_{\ell^{(n')}} \leq O(\log n') \max_{x \in \mathbb{S}^{n'-1}} \|x^{B_{j^*}}\|_\ell \leq O(\log n') \max_{x \in \mathbb{S}^{n'-1}} \|x^{B_{j^*}}\|_2 \cdot \left\| \frac{x^{B_{j^*}}}{\|x^{B_{j^*}}\|_2} \right\|_\ell$$

$$\leq O(\log n') \max_{x \in \mathbb{S}^{n'-1}} \left\| \frac{x^{B_j}}{\|x^{B_j}\|_2} \right\|_\ell \leq O(\log n') \max_{b_{j*} \leq n'} \|\xi^{(b_j)}\|_{\ell^{(b_j)}} \leq O(\log n') \max_{b_{j*} \leq n'} \frac{1}{\overline{G}(b_{j*})} \leq \frac{O(\sqrt{C_G} \log n')}{\overline{G}(n')}.$$

Thus, we have

$$\operatorname{mmc}(\ell) = \max_{n' \in [n]} \frac{\max_{x \in \mathbb{S}^{n'-1}} \|x\|_{\ell^{(n')}}}{M_{\ell^{(n')}}} \leq O(\sqrt{C_G} \log n).$$

$\square$

## C.4 Contraction Bound of SymSketch

In this section we give the contraction bound of SymSketch. We first show that for a fixed vector $x \in \mathbb{R}^n$, $\|\widetilde{D}x\|_2 \geq 1/\operatorname{poly}(d \log n) \cdot \|x\|_\ell$ with probability $1 - 2^{-\Theta(d \log n)}$.

**Lemma 26.** *Let $\widetilde{D}$ be the matrix defined in Definition 5. For any fixed $x \in \mathbb{R}^n$, with probability $1 - 2^{-\Theta(d \log n)}$, $\|\widetilde{D}x\|_2 \geq 1/\alpha_0 \cdot \|x\|_\ell$, where $\alpha_0 = O(\operatorname{mma}(\ell, d) \cdot \log^{5/2} n) = O(\sqrt{d} \log^3 n)$.*

*Proof.* The lemma follows from the following two claims. Recall that $t = \Theta(\log n)$.

**Claim 1.** *For any fixed $x \in \mathbb{R}^n$. If there is a contributing level $i^* \in \{0, 1, 2, \ldots, g\}$ such that $|L_{i^*}(x)| = \Theta(d \log n) \cdot 2^j$ for some $j \in [t]$, then with probability at least $1 - 2^{-\Theta(d \log n)}$,*

$$\|w_j D_j x\|_2 \geq \Omega \left( \frac{1}{\operatorname{mma}(\ell, d) \cdot \log^{5/2} n} \right) \|x\|_\ell.$$

*Proof.* Let $y_h$ be a random variable such that

$$y_h = \begin{cases} 1 & \text{if the } h\text{-th diagonal entry of } D_j \text{ is } 1 \\ 0 & \text{otherwise} \end{cases}.$$

Let $Y = \sum_{h \in L_{i^*}(x)} y_h$. By Chernoff bound, we have

$$\Pr[Y \geq \Omega(d \log n)] \geq 1 - 2^{-\Theta(d \log n)}.$$

Conditioned on $Y \geq \Omega(d \log n)$, we have

$$
\begin{aligned}
\frac{\|w_j D_j x\|_2}{\|x\|_\ell} &= \frac{w_j \|D_j x\|_2}{\|x\|_\ell} \\
&\geq \frac{2^{i^*} w_j \sqrt{d \log n}}{\|x\|_\ell} \\
&\geq \frac{2^{i^*} \sqrt{2^j} M_{\ell(2^j)} \sqrt{d \log n}}{2^{i^*+1} g M_{\ell(|L_{i^*}(x)|)} \sqrt{\log n} \sqrt{|L_{i^*}(x)|}} \\
&\geq \Omega(1/\log^{3/2} n) \cdot \frac{M_{\ell(2^j)}}{M_{\ell(|L_{i^*}(x)|)}} \\
&= \Omega(1/\log^{3/2} n) \cdot \frac{M_{\ell(2^j)}}{M_{\ell(|L_{i^*}(x)|/\log n)}} \cdot \frac{M_{\ell(|L_{i^*}(x)|/\log n)}}{M_{\ell(|L_{i^*}(x)|)}} \\
&\geq \Omega \left( \frac{1}{\operatorname{mma}(\ell, d) \cdot \operatorname{mma}(\ell, \log n) \cdot \log^{3/2} n} \right) \\
&\geq \Omega \left( \frac{1}{\operatorname{mma}(\ell, d) \cdot \log^{5/2} n} \right).
\end{aligned}
$$

Here the first inequality follows from the fact that there are at least $\Omega(d \log n)$ coordinates sampled from $L_{i^*}(x)$. The second inequality follows from Lemma 20 and the fact that level $i^*$ is a contributing level. The third inequality follows from $|L_{i^*}(x)| = \Theta(d \log n) \cdot 2^j$ and $g = \Theta(\log n)$. The forth inequality follows from Definition 13. The last inequality follows from Lemma 21. $\qquad \square$

**Claim 2.** *For any fixed $x \in \mathbb{R}^n$. If there is a contributing level $i^* \in \{0, 1, 2, \ldots, g\}$ such that $|L_{i^*}(x)| = O(d \log n)$, then we have*

$$\|w_0 D_0 x\|_2 \geq \Omega \left( \frac{1}{\operatorname{mma}(\ell, d) \cdot \log^{5/2} n} \right) \|x\|_\ell.$$

*Proof.*

$$\frac{\|w_0 D_0 x\|_2}{\|x\|_\ell} = \frac{w_0 \|x\|_2}{\|x\|_\ell}$$

$$\geq \frac{2^{i^*} w_0 \sqrt{|L_{i^*}(x)|}}{\|x\|_\ell}$$

$$\geq \frac{2^{i^*} M_{\ell^{(1)}} \sqrt{|L_{i^*}(x)|}}{2^{i^*+1} g M_{\ell(|L_{i^*}(x)|)} \sqrt{\log n} \sqrt{|L_{i^*}(x)|}}$$

$$\geq \Omega(1/\log^{3/2} n) \cdot \frac{M_{\ell^{(1)}}}{M_{\ell(|L_{i^*}(x)|/\log n)}} \cdot \frac{M_{\ell(|L_{i^*}(x)|/\log n)}}{M_{\ell(|L_{i^*}(x)|)}}$$

$$\geq \Omega\left(\frac{1}{\mathrm{mma}(\ell, d) \cdot \log^{5/2} n}\right)$$

The first inequality follows from the fact that we only consider the contribution of the coordinates in $L_{i^*}(x)$. The second inequality follows from Lemma 20 and the fact that level $i^*$ is a contributing level. The third inequality follows from $g = \Theta(\log n)$. The last inequality follows from Definition 13 and Lemma 21. $\qquad\square$

By Claim 1 and Claim 2, since any vector $x \in \mathbb{R}^n$ contains at least one contributing level, with probability at least $1 - 2^{-\Theta(d \log n)}$ we have $\|\widetilde{D}x\|_2 \geq \Omega(1/(\mathrm{mma}(\ell, d) \cdot \log^{5/2} n)) \cdot \|x\|_\ell$. We complete the proof by combining this with Lemma 21. $\qquad\square$

Now we show how to combine the contraction bound in Lemma 26 with a net argument to give a contraction bound for all vectors in a subspace.

**Lemma 27.** *Let $S \in \mathbb{R}^{O(d) \times n}$ be a random matrix. For any $\alpha_0 = \mathrm{poly}(n)$ and $A \in \mathbb{R}^{n \times d}$, if*

1. *$\|S\|_2 \leq \mathrm{poly}(n)$ holds with probability at least $0.999$;*

2. *for any fixed $x \in \mathbb{R}^n$, $\|Sx\|_2 \geq 1/\alpha_0 \cdot \|x\|_\ell$ holds with probability $1 - e^{-Cd \log n}$ for a sufficiently large constant $C$,*

*then with probability at least $0.998$, for all $y \in \mathbb{R}^n$ in the column space of $A$,*
$$\|Sy\|_2 \geq \Omega(1/\alpha_0)\|y\|_\ell.$$

*Proof.* For the matrix $A \in \mathbb{R}^{n \times d}$, we define the set $\mathcal{B} = \{y \mid y = Ax, \|y\|_2 = 1\}$. We define $\mathcal{N} \subset \mathbb{R}^n$ to be an $\varepsilon$-net of $\mathcal{B}$ as in Definition 8. By Lemma 13, we have $|\mathcal{N}| \leq (1 + 1/\varepsilon)^d$, and for all $y \in \mathcal{B}$, there exists $z \in \mathcal{N}$ such that $\|y - z\|_2 \leq \varepsilon$. We take $\varepsilon = 1/\mathrm{poly}(n)$ here.

Due to the second condition, since $|\mathcal{N}| \leq e^{O(d \log n)}$, by taking union bound over all vectors in $\mathcal{N}$, we know that with probability $1 - e^{-\Theta(d \log n)}$, for all $z \in \mathcal{N}$, $\|Sz\|_2 \geq 1/\alpha_0 \cdot \|z\|_\ell$.

Now, for any vector $y \in \mathcal{B}$, there exists $z \in \mathcal{N}$ such that $\|y - z\|_2 \leq 1/\mathrm{poly}(n)$, and we define $w = y - z$.

$$
\begin{aligned}
\|Sy\|_2 &= \|S(z + w)\|_2 \\
&\geq \|Sz\|_2 - \|Sw\|_2 && \text{(triangle inequality)} \\
&\geq 1/\alpha_0 \cdot \|z\|_\ell - \|Sw\|_2 && \text{(by the second condition)} \\
&\geq 1/\alpha_0 \cdot \|z\|_\ell - \|S\|_2 \|w\|_2 && (\|Sw\|_2 \leq \|S\|_2 \cdot \|w\|_2) \\
&\geq 1/\alpha_0 \cdot \|z\|_\ell - \mathrm{poly}(n) \cdot \|w\|_2 && \text{(by the first condition)} \\
&\geq 1/\alpha_0 \cdot \|y - w\|_\ell - \mathrm{poly}(n) \cdot \|w\|_2 && (y = z + w) \\
&\geq 1/\alpha_0 \cdot \|y\|_\ell - 1/\alpha_0 \cdot \|w\|_\ell - \mathrm{poly}(n) \cdot \|w\|_2 && \text{(triangle inequality)} \\
&\geq 1/\alpha_0 \cdot \|y\|_\ell - 1/\alpha_0 \cdot \sqrt{n}\|w\|_2 - \mathrm{poly}(n) \cdot \|w\|_2 && \text{(Lemma 19)} \\
&\geq 1/\alpha_0 \cdot \|y\|_\ell - (1/\alpha_0 \cdot \sqrt{n} + \mathrm{poly}(n))\varepsilon && (\|w\|_2 \leq \varepsilon) \\
&\geq 0.5/\alpha_0 \cdot \|y\|_\ell.
\end{aligned}
$$

$\qquad\square$

**Lemma 28.** *For a given matrix $A \in \mathbb{R}^{n \times d}$. Let $S \in \mathbb{R}^{O(d) \times n}$ be a $\mathsf{SymSketch}$ as defined in Definition 5. With probability at least $0.995$, for all $x \in \mathbb{R}^d$, $\|SAx\|_2 \geq 1/\alpha_0 \cdot \|Ax\|_\ell$ where $\alpha_0 = O(\sqrt{d} \log^3 n)$.*

*Proof.* By Lemma 26 and Lemma 23, the two conditions in Lemma 27 are satisfied. By Lemma 27, with probability at least $0.998$, for all $x \in \mathbb{R}^d$, $\|\widetilde{D}Ax\|_2 \geq \Omega(1/\alpha_0)\|Ax\|_\ell$. Since $\Pi \in \mathbb{R}^{O(d) \times n(t+1)}$ is a composition of Gaussian embedding and $\mathsf{CountSketch}$ embedding with $\varepsilon = 0.1$, by Corollary 17, with probability at least $0.999$, for all $x \in \mathbb{R}^d$, $\|\Pi\widetilde{D}Ax\|_2 \geq \Omega(\|\widetilde{D}Ax\|_2)$. By a union bound, we know that with probability at least $0.995$, for all $x \in \mathbb{R}^d$, $\|SAx\|_2 \geq \Omega(1/\alpha_0)\|Ax\|_\ell$. $\qquad \square$

## C.5 Dilation Bound of $\mathsf{SymSketch}$

In this section we give the dilation bound of $\mathsf{SymSketch}$. We first show that for any fixed $x \in \mathbb{R}^n$, with high probability, $\|\widetilde{D}x\|_2 \leq \mathrm{poly}(d \log n) \cdot \mathrm{mmc}(\ell) \cdot \|x\|_\ell$.

**Lemma 29.** *Let $\widetilde{D}$ be the matrix defined in Definition 5. For any fixed vector $x \in \mathbb{R}^n$, with probability $1 - \delta$, $\|\widetilde{D}x\|_2 \leq \alpha_1/\delta \cdot \|x\|_\ell$, where $\alpha_1 = O(\mathrm{mmc}(\ell) \log^{5/2} n)$.*

*Proof.* Consider a fixed vector $x \in \mathbb{R}^n$. Recall that $t = \Theta(\log n)$. Let $c > 0$ be a fixed constant. We define the $j$-heavy level set $H_j$ as

$$H_j = \left\{ i \;\middle|\; |L_i(x)| \geq c\frac{\delta 2^j}{\log^2 n}, 0 \leq i \leq g \right\}.$$

Let $\overline{H}_j$ be the $j$-light level set, i.e., $\overline{H}_j = \{0, 1, \ldots, g\} \setminus H_j$. Notice that

$$\sum_{i \in \overline{H}_j} |L_i(x)| \cdot 2^{-j} \leq g \cdot c\delta 2^j / \log^2 n \cdot 2^{-j} \leq O(\delta / \log n).$$

By Markov's inequality, with probability at least $1 - \delta/(2t)$, no element from a $j$-light level is sampled by $D_j$, i.e., for all $i \in \overline{H}_j, k \in L_i(x)$, the $k$-th diagonal entry of $D_j$ is $0$. By taking union bound over all $j \in [t]$, with probability at least $1 - \delta/2$, for all $j \in [t]$, no element from a $j$-light level is sampled by $D_j$. Let $\zeta$ denote this event. We condition on this event in the remaining part of the proof. In the following analysis, we show an upper bound of $\|w_j D_j x\|_2^2$ for each $j \in [t]$. Let $H_j$ be the set of $j$-heavy levels.

Consider a fixed $j \in [t]$, we have

$$\begin{aligned}
\mathop{\mathbf{E}}_{D_j}\left[\|w_j D_j x\|_2^2 \;\middle|\; \zeta\right] &= w_j^2 \mathop{\mathbf{E}}_{D_j}\left[\|D_j x\|_2^2 \;\middle|\; \zeta\right] \\
&= w_j^2 \mathop{\mathbf{E}}_{D_j}\left[\sum_{h=1}^{n} (D_j(h,h))^2 x_h^2 \;\middle|\; \zeta\right] \\
&= w_j^2 \mathop{\mathbf{E}}_{D_j}\left[\sum_{i=0}^{g} \sum_{h \in L_i(x)} (D_j(h,h))^2 x_h^2 \;\middle|\; \zeta\right] \\
&= w_j^2 \frac{1}{2^j} \sum_{i \in H_j} \sum_{h \in L_i(x)} x_h^2 \\
&\leq w_j^2 \frac{1}{2^j} \sum_{i \in H_j} |L_i(x)| \cdot (2^{i+1})^2.
\end{aligned}$$

**Claim 3.** $w_j^2 2^{-j} \leq O((M_{\ell(2^j)})^2)$.

*Proof.*

$$w_j^2 2^{-j} = (\|(1, 1, \ldots, 1, 0, \ldots, 0)\|_\ell)^2 \cdot 2^{-j}$$

$$= \left( \left\| \frac{1}{\sqrt{2^j}} (1, 1, \ldots, 1, 0, \ldots, 0) \right\|_\ell \sqrt{2^j} \right)^2 \cdot 2^{-j}$$

$$= (\|\xi^{(2^j)}\|_\ell)^2 \cdot 2^j \cdot 2^{-j}$$

$$= (\|\xi^{(2^j)}\|_\ell)^2$$

$$\leq O(M_{\ell(2^j)}),$$

where the third step follows from the definition of $\xi^{(2^j)}$, and the last step follows from Lemma 20. $\square$

Using the above claim, we have

$$w_j^2 \sum_{i \in H_j} |L_i(x)| \cdot 2^{2i-j}$$

$$= \sum_{i \in H_j} \frac{w_j^2 2^{-j}}{(M_{\ell(|L_i(x)|)})^2} \cdot |L_i(x)| \cdot (M_{\ell(|L_i(x)|)})^2 \cdot 2^{2i}$$

$$\leq O\left( \sum_{i \in H_j} \left( \frac{M_{\ell(2^j)}}{M_{\ell(|L_i(x)|)}} \right)^2 \cdot |L_i(x)| \cdot (M_{\ell(|L_i(x)|)})^2 \cdot 2^{2i} \right)$$

$$\leq O\left( \sum_{i \in H_j} \left( \frac{M_{\ell(2^j)}}{M_{\ell(|L_i(x)|)}} \right)^2 w_{\log |L_i(x)|}^2 \log n \cdot 2^{2i} \right)$$

$$= \log n \cdot \underbrace{\sum_{i \in H_j, |L_i(x)| \leq 2^j} \left( \frac{M_{\ell(2^j)}}{M_{\ell(|L_i(x)|)}} \right)^2 \cdot w_{\log |L_i(x)|}^2 \cdot 2^{2i}}_{\diamond}$$

$$+ \log n \cdot \underbrace{\sum_{i \in H_j, |L_i(x)| > 2^j} \left( \frac{M_{\ell(2^j)}}{M_{\ell(|L_i(x)|)}} \right)^2 \cdot w_{\log |L_i(x)|}^2 \cdot 2^{2i}}_{\heartsuit},$$

where the second step follows from $w_j^2 2^{-j} \leq O((M_{\ell(2^j)})^2)$ (Claim 3), and the third step follows from $|L_i(x)| \cdot (M_{\ell(|L_i(x)|)})^2 \leq O(w_{\log |L_i(x)|}^2 \log n)$ (Lemma 20). It remains to upper bound $\diamond$ and $\heartsuit$.

To given an upper bound for $\diamond$, we have

$$\diamond \leq O\left( \sum_{i \in H_j, |L_i(x)| \leq 2^j} \mathrm{mma}^2(\ell, \log^2 n/\delta) \cdot w_{\log |L_i(x)|}^2 \cdot 2^{2i} \right)$$

$$\leq O\left( \mathrm{mma}^2(\ell, \log^2 n/\delta) \left( \sum_{i=0}^{g} w_{\log |L_i(x)|} \cdot 2^i \right)^2 \right)$$

$$\leq O\left( \mathrm{mma}^2(\ell, \log^2 n/\delta) \right) \|x\|_\ell^2$$

$$\leq O(\log^3 n/\delta) \|x\|_\ell^2,$$

where the first step follows from the definition of $\mathrm{mma}$, the second step follows from Minkowski inequality, the third step follows from the definition of $L_i(x)$, $w_{\log |L_i(x)|}$ and triangle inequality, the last step follows from Lemma 21.

To give an upper bound for $\heartsuit$, we have

$$\heartsuit \le O\left( \log n \cdot \sum_{i \in H_j, |L_i(x)| > 2^j} \mathrm{mmc}^2(\ell) w_{\log |L_i(x)|}^2 \cdot 2^{2i} \right)$$

$$\le O\left( \log n \cdot \mathrm{mmc}^2(\ell) \cdot \left( \sum_{i=0}^{g} w_{\log |L_i(x)|} \cdot 2^i \right)^2 \right)$$

$$\le O\left( \log n \cdot \mathrm{mmc}^2(\ell) \cdot \|x\|_\ell^2 \right),$$

where the first step follows from $(M_{\ell(2^j)}/M_{\ell(|L_i(x)|)})^2 \le O(\log n \cdot \mathrm{mmc}^2(\ell))$ (Lemma 3.14 in [7]). Putting it all together, we have

$$\mathop{\mathbf{E}}_{D_j}[\|w_j D_j x\|_2^2 | \zeta] \le \log n \cdot (\diamondsuit + \heartsuit) \le O(\log^4 n/\delta + \log^2 n \cdot \mathrm{mmc}^2(\ell)) \|x\|_\ell^2.$$

Thus,

$$\mathop{\mathbf{E}}_{\widetilde{D}}[\|\widetilde{D} x\|_2^2 | \zeta] \le \sum_{j=0}^{t} \mathop{\mathbf{E}}_{D_j}[\|w_j D_j x\|_2^2 | \zeta] \le O(\log^5 n/\delta + \log^3 n \cdot \mathrm{mmc}^2(\ell)) \|x\|_\ell^2.$$

By Markov's inequality, conditioned on $\zeta$, with probability at least $1 - \delta/2$,

$$\|\widetilde{D} x\|_2^2 \le O(\log^5 n/\delta + \log^3 n \cdot \mathrm{mmc}^2(\ell)) \|x\|_\ell^2/\delta.$$

Since $\zeta$ holds with probability at least $1 - \delta/2$, with probability at least $1 - \delta$, we have

$$\|\widetilde{D} x\|_2 \le O(\log^{5/2} n/\delta \cdot \mathrm{mmc}(\ell)) \cdot \|x\|_\ell.$$

$\square$

Now we show how to use the dilation bound for a fixed vector in Lemma 29 to prove a dilation bound for all vectors in a subspace. We need the following existential result in our proof.

**Lemma 30** ([6]). *Given a matrix $A \in \mathbb{R}^{n \times m}$ and a norm $\|\cdot\|$, there exists a basis matrix $U \in \mathbb{R}^{n \times d}$ of the column space of $A$, such that*

$$\sum_{i=1}^{d} \|U^i\| \le d,$$

*and for all $x \in \mathbb{R}^d$,*

$$\|x\|_\infty \le \|Ux\|.$$

**Lemma 31.** *Given a matrix $A \in \mathbb{R}^{n \times d}$. Let $S \in \mathbb{R}^{O(d) \times n}$ be a SymSketch as defined in Definition 5. With probability at least $0.99$, for all $x \in \mathbb{R}^d$,*

$$\|SAx\|_2 \le O(\alpha_1 d^2) \|Ax\|_\ell,$$

*where $\alpha_1 = O(\mathrm{mmc}(\ell) \cdot \log^{5/2} n)$.*

*Proof.* Recall that $S = \Pi \widetilde{D}$. Let $U$ be a basis matrix of the column space of $A$ as in Lemma 30. By Lemma 29, for a fixed $i \in [d]$, with probability at least $1 - 1/(100d)$, $\|\widetilde{D} U^i\|_2 \le O(\alpha_1 d) \|U^i\|_\ell$. By taking a union bound over $i \in [d]$, with probability at least $0.999$, for all $i \in [d]$, $\|\widetilde{D} U^i\|_2 \le$

$\alpha_1 d \|U^i\|_\ell$. Thus, for any $x \in \mathbb{R}^d$,

$$
\begin{aligned}
\|\widetilde{D}Ux\|_2 &\leq \sum_{i=1}^{d} |x_i| \cdot \|\widetilde{D}U^i\|_2 \\
&\leq \|x\|_\infty \cdot \sum_{i=1}^{d} \|\widetilde{D}U^i\|_2 \\
&\leq \|Ux\|_\ell \cdot \sum_{i=1}^{d} \|\widetilde{D}U^i\|_2 \\
&\leq O(\alpha_1 d) \cdot \|Ux\|_\ell \cdot \sum_{i=1}^{d} \|U^i\|_\ell \\
&\leq O(\alpha_1 d^2) \cdot \|Ux\|_\ell,
\end{aligned}
$$

where the first step follows from triangle inequality, the second step follows from $|x_i| \leq \|x\|_\infty$ for all $i \in [d]$, the third step follows from $\|x\|_\infty \leq \|Ux\|_\ell$, the fourth step follows from $\|\widetilde{D}U^i\|_2 \leq O(\alpha_1 d)\|U^i\|_\ell$, the last step follows from $\sum_{i=1}^{d} \|U^i\|_\ell \leq d$.

By Corollary 17, with probability at least 0.999, $\Pi$ is an $\ell_2$ subspace embedding with $\varepsilon = 0.1$ for the column space of $\widetilde{D}U$. Thus, with probability at least 0.99, for all $x \in \mathbb{R}^d$, $\|SAx\|_2 \leq O(\alpha_1 d^2)\|Ax\|_\ell$. □

### C.6 Proof of Theorem 11

*Proof.* It directly follows from Lemma 28, Lemma 31 and Lemma 22. □

## D Missing Proofs of Main Theorems

### D.1 Proof of Theorem 1

Let $S \in \mathbb{R}^{O(d) \times n}$ be a SymSketch as defined in Definition 5, and $\Pi = O(\sqrt{d}\log^3 n) \cdot S$. By Corollary 12, for a given matrix $A \in \mathbb{R}^{n \times d}$, with probability at least 0.9, for all $x \in \mathbb{R}^d$,

$$\|Ax\|_G \leq \|\Pi Ax\|_2 \leq \kappa \|Ax\|_G,$$

where $\kappa = O(\sqrt{C_G} d^{5/2} \log^{13/2} n)$. We prove Theorem 1 by combining Theorem 8 with the embedding matrix $\Pi$ constructed above.

### D.2 Proof of Theorem 2

Let $S \in \mathbb{R}^{O(d) \times n}$ be a SymSketch as defined in Definition 5. For a given data matrix $A \in \mathbb{R}^{n \times d}$ and response vector $b \in \mathbb{R}^n$, we calculate $x^* = \operatorname{argmin}_x \|SAx - Sb\|_2$ and return $x^*$. The algorithm runs in $O(\mathrm{nnz}(A) + \mathrm{poly}(d))$ time, since by Lemma 22, the expected running time for calculating $SA$ is $O(\mathrm{nnz}(A) + \mathrm{poly}(d))$, and $x^* = (SA)^+ Sb$ can be calculated in $\mathrm{poly}(d)$ time.

To see the correctness, let $\overline{x} = \operatorname{argmin}_x \|Ax - b\|_\ell$. With probability at least 0.99, we have

$$
\begin{aligned}
\|Ax^* - b\|_\ell &\leq O(\sqrt{d}\log^3 n)\|SAx^* - Sb\|_2 \\
&\leq O(\sqrt{d}\log^3 n)\|SA\overline{x} - Sb\|_2 \\
&\leq O(\sqrt{d}\log^3 n)\|\widetilde{D}A\overline{x} - \widetilde{D}b\|_\ell \\
&= O(\sqrt{d}\log^{11/2} n) \cdot \mathrm{mmc}(\ell) \cdot \|A\overline{x} - b\|_\ell.
\end{aligned}
$$

The first step follows by applying Lemma 28 on $\overline{A}$, where we use $\overline{A} \in \mathbb{R}^{n \times (d+1)}$ to denote a matrix whose first $d$ columns are $A$ and the last column is $b$. The second step follows from the fact that $x^* = \operatorname{argmin}_x \|SAx - Sb\|_2$. The third step follows by Definition 5 and Corollary 17. The last step follows by applying Lemma 29 on $A\overline{x} - b$.

Figure 2: Experiments on Orlicz norm.

# E    Experiments

In this section, we perform experiments to validate the practicality of our methods.

**Experiment Setup.**    We compare the proposed algorithms with baseline algorithms on the U.S. 2000 Census Data containing $n = 5 \times 10^6$ rows and $d = 11$ columns and UCI YearPredictionMSD dataset which has $n = 515,345$ rows and $d = 90$ columns. All algorithms are implemented in Python 3.7. To solve the optimization problems induced by the regression problems and their sketched versions, we invoke the `minimize` function in `scipy.optimize`. Each experiment is repeated for *25 times*, and the mean of the loss function value is reported. In all experiments, we vary the sampling size or embedding dimension from $5d$ to $20d$, and observe their effects on the quality of approximation.

**Experiments on Orlicz Norm.**    We compare our algorithm in Section 2 with uniform sampling and the embedding in [2]. We also calculate the optimal solution to verify the approximation ratio. We try Orlicz norms induced by two different $G$ functions: Huber with $c = 0.1$ and "$\ell_1 - \ell_2$". See Table 1 for definitions. Our experimental results in Figure 2 clearly demonstrate the practicality of our algorithm. In both datasets, our algorithm outperforms both baseline algorithms by a significant margin, and achieves the best accuracy in almost all settings.

**Experiments on Symmetric Norm.**    We compare our algorithm in Section 3 (SymSketch) with the optimal solution to verify the approximation ratio. We try two different symmetric norms: top-$k$ norm with $k = n/5$ and sum-mix of $\ell_1$ and $\ell_2$ norm ($\|x\|_1 + \|x\|_2$). As shown in Figure 3, SymSketch achieves reasonable approximation ratios with moderate embedding dimension. In particular, the algorithm achieves an approximation ratio of $1.25$ when the embedding dimension is only $5d$.

Figure 3: Experiments on symmetric norm.