[Reviews · NeurIPS 2019]

Reviewer 1



The paper considers the problem of sketching in linear regression with symmetric norm loss functions. They provide a novel sketching method for generic symmetric norms that can provide a $d\text{polylog}n$ approximate to the underlying solution. This is an interesting result especially due to the fact that it generalizes the previous results in the literature. However, unlike other results in the literature (which mostly focus on $\ell_p$-norms and recently M-estimators), it can not provide a $1+\epsilon$-estimate of the solution (this is presented as a future direction in the conclusion section.) For the special case of Orlicz norm, the row sampling algorithm according to the Orlicz norm leverage score is a novel algorithm that can improve the previous results (in [2]). However, it should be emphasized that the Assumption 1. limits the applicability of the result. Specifically, due to the fact that the growth in function $G$ is sub-quadratic, this algorithm can not be used when the loss function is an $\ell_p$-norm with $p>2$. I understand that this a theory paper. However, in order to show the applicability of the algorithms I expected to see some experimental results to compare the time complexity and the quality of the estimate of the proposed algorithm with the previous results in the literature. The paper lacks providing experimental results which would have been very interesting for NeurIPS audience.

Reviewer 2



After reading the response, I tend to agree with the authors that the "Orlicz norm leverage scores" are a nice contribution. However, I would still appreciate a more detailed discussion and comparison between their results and Clarkson and Woodruff (2015) [11]. -------------- The authors present new sketching algorithms for linear regression under a family of (nice) Orlicz norms, and also more generally, symmetric norms. All of the algorithms run in O(nnz(A)+poly(d)) time, where nnz(A) is the number of nonzeros in the n x d input matrix A (whose rows represent data points in the regression). In the case of Orlicz norms, the proposed algorithm obtains an eps-approximation for the loss of the optimum, and in the case of arbitrary symmetric norm, the approximation quality is sqrt(d) x polylog(n) x mmc(l), where mmc(l) is a sort of condition number of the norm l. The prior works on this obtained a d x polylog n approximation for Orlicz norms and an eps-approximation for a related class of M-estimators and a narrower class of l_p norms. The authors mention a couple of examples of symmetric and Orlicz norms that do not fall under those other classes of regression losses. To my understanding, the contribution is of a primarily theoretical interest. Thus it is important to take into account the novelty of the proofs. I’m not expert and did not go over the proofs in the appendix in significant detail. My overall impression is that there are two contributions: 1. A subspace embedding sketch for symmetric norms, which is an incremental improvement over the prior work, which used similar sketches in a somewhat different data streaming context [7]. 2. Analysis of sampling based on “Orlicz norm leverage scores” which is described as the primary contribution. These techniques look somewhat similar to the “leverage score sampling” used in Clarkson and Woodruff (2015) [11]. Even though [11] uses different leverage scores (based on l_p norms, I believe) and they claim guarantees for M-estimators rather than Orlicz norms, my impression is that those techniques may apply here as well (I refer to Section 4.2 of the arxiv version of [11]). One argument for this is that, even though Orlicz norms are not "entry-wise" norms, the analysis in this paper immediately makes a transformation that converts the norms into "entry-wise" expressions that are very similar to M-estimators (see equation (6) and proof of Lemma 7). As I said, I’m not an expert, but even if the authors disagree, this would be worth discussing more closely in the paper. Minor comments: - Line 33: "Despite we have successfully applied" -> Despited having successfully applied - Line 40: "Is that possible" -> Is it possible - Line 43: "a list a M-estimators" -> a list of M-estimators - Line 55: "is prohibited for" -> is prohibitive for - Line 76: "function G appeared in" -> function G which appeared in - Line 96: "as follow" -> as follows - Line 221,223: "least square" -> least squares

Reviewer 3



The definition in (2) seems quite important for the analysis, e.g., the proof of Lemma 7. But the original definition of Orlicz norm is slightly different from (2). That is, in (2), the equality is strictly hold and the $¥alpha$ should be unique. I'm not sure the whether (2) and the definition of the Orlicz norm is equivalent or not. Please make sure the relationship between these two. The authors mainly focus of the regression problem in the article. However, classification or non-linear regression are also important in practice. Could the algorithm extend to such a more general problem?

[Author Response · NeurIPS 2019]

We thank all reviewers for their comments, and will incorporate suggestions in the final version. Although the goal of
this paper is theoretical, we perform experiments to resolve reviewers' concern about practicality of our methods.
**Experiment Setup.** We compare the proposed algorithms with baseline algorithms on the U.S. 2000 Census Data
containing $n = 5 \times 10^6$ rows and $d = 11$ columns and UCI YearPredictionMSD dataset which has $n = 515,345$ rows
and $d = 90$ columns. All algorithms are implemented in Python 3.7. To solve the optimization problems induced by
the regression problems and their sketched versions, we invoke the `minimize` function in `scipy.optimize`. Each
experiment is repeated for *25 times*, and the mean of the loss function value is reported. In all experiments, we vary the
sampling size or embedding dimension from $5d$ to $20d$, and observe their effects on the quality of approximation.
**Experiments on Orlicz norm.** We compare our algorithm in Section 2 with uniform sampling and the embedding
in [2]. We also calculate the optimal solution to verify the approximation ratio. We try Orlicz norms induced by
two different $G$ functions: Huber with $c = 0.1$ and "$\ell_1 - \ell_2$". See Table 1 in our submission for definitions. Our
experimental results given below clearly demonstrate the practicality of our algorithm. In both datasets, our algorithm
   outperforms both baseline algorithms by a significant margin, and achieves the best accuracy in almost all settings.

**Experiments on symmetric norm.** We compare our algorithm in Section 3 (SymSketch) with the optimal solution to
verify the approximation ratio. We try two different symmetric norms: top-$k$ norm with $k = n/5$ and sum-mix of $\ell_1$ and
$\ell_2$ norm ($\|x\|_1 + \|x\|_2$). See Line 58-60 in our submission for definitions of these norms. As shown below, SymSketch
achieves reasonable approximation ratios with moderate embedding dimension. In particular, the algorithm achieves an
   approximation ratio of $1.25$ when the embedding dimension is only $5d$.

**(Reviewer #1) Assumption 1.** Our sampling algorithm in fact works for $\ell_p$ norms when $p > 2$. In general, suppose
the function $G : \mathbb{R} \to \mathbb{R}$ satisfies that for all $0 < x < y$, $G(y)/G(x) \le C_G(y/x)^p$, for the Orlicz norm induced by
$G$, given a well-conditioned basis with condition number $\kappa_G$, our sampling algorithm returns a matrix with roughly
$O((\sqrt{d}\kappa_G)^p \cdot d/\varepsilon^2)$ rows such that Theorem 1 holds. However it is not clear how to calculate well-conditioned bases
in input-sparsity time when $p > 2$. Our current method fails since it requires an oblivious subspace with $\text{poly}(d)$
distortion, and it is known that such embedding does not exist when $p > 2$ [9]. Since we focus on input-sparsity time
algorithms in this paper, we did not include the $p > 2$ case. We will add more discussion on this in the final version.
**(Reviewer #2) Results on symmetric norm.** We disagree that this is an incremental improvement. First, the previous
embedding with $d \log n$ distortion only works for Orlicz norms, and in this paper we give the first subspace embedding
for general symmetric norms. Second, the construction in [7] is only for streaming algorithms. To construct a subspace
embedding, we need to show that (i) norms of all vectors in a subspace are preserved and (ii) there is a simple estimator
in the sketch space. Neither of them can be satisfied by the construction in [7].
**Comparison with [11].** First, our definitions for Orlicz norm leverage score and well-conditioned basis, as given in
Definition 2 and 3, are different from all previous works and are closely related to the Orlicz norm under consideration.
The algorithm in [11], on the other hand, simply uses $\ell_p$ leverage scores. Under our definition, we can prove that the
sum of leverage scores is bounded by $O(C_G d\kappa_G^2)$ (Lemma 4), whose proof requires a novel probabilistic argument. In
contrast, the upper bound on sum of leverage scores in [11] is $O(\sqrt{nd})$ (Lemma 38 in [11]). Thus, the algorithm in
[11] runs in an iterative manner since in each round the algorithm can merely reduce the dimension from $n$ to $O(\sqrt{nd})$,
while our algorithm is one-shot. We will of course add a more detailed comparison with [11] in the final version.
**(Reviewer #3)** The uniqueness of $\alpha$ follows from the assumption that $G$ is strictly increasing, in which case the two
definitions are equivalent. The assumption that $G$ is strictly increasing was also implicitly made in Andoni et al. [2].
It is indeed an interesting problem to generalize our techniques to other problems, e.g., classification problems and
non-linear regression problems. We leave this as a future work.

[Meta-Review · NeurIPS 2019]

This is a technically sound paper on the efficient solving of linear regression problems posed wrt symmetric norms, where efficiency is obtained thanks to a clever random embedding of the data that preserves the (symmetric) norm. It might be nice for the final version of the paper to include in the main text empirical evidence of the proposed theoretical results.